# Adipose tissue–specific ablation of Ces1d causes metabolic dysregulation in mice

Gang Li[1,*], Xin Li[1,*], Li Yang[1], Shuyue Wang[1], Yulin Dai[2], Baharan Fekry[1], Lucas Veillon[3], Lin Tan[3], Rebecca Berdeaux[1,4,5], Kristin Eckel-Mahan[1,4,5], Philip L Lorenzi[3], Zhongming Zhao[2], Richard Lehner[6], Kai Sun[1,4,5]

**Carboxylesterase 1d (Ces1d) is a crucial enzyme with a wide range of activities in multiple tissues. It has been reported to localize predominantly in ER. Here, we found that Ces1d levels are significantly increased in obese patients with type 2 diabetes. Intriguingly, a high level of Ces1d translocates onto lipid droplets where it digests the lipids to produce a unique set of fatty acids. We further revealed that adipose tissue–specific Ces1d knock-out (FKO) mice gained more body weight with increased fat mass during a high fat-diet challenge. The FKO mice exhibited impaired glucose and lipid metabolism and developed exacerbated liver steatosis. Mechanistically, deficiency of Ces1d induced abnormally large lipid droplet deposition in the adipocytes, causing ectopic accumulation of triglycerides in other peripheral tissues. Furthermore, loss of Ces1d diminished the circulating free fatty acids serving as signaling molecules to trigger the epigenetic regulations of energy metabolism via lipid-sensing transcriptional factors, such as HNF4α. The metabolic disorders induced an unhealthy microenvironment in the metabolically active tissues, ultimately leading to systemic insulin resistance.**

## Introduction

Obesity is a pathological change majorly caused by abnormal accumulation of body fat mass, which indicates the failure of the body to ensure the proper energy homeostasis (1, 2). Adipose tissue is the predominant site to buffer the energy in neutral lipid form. During the development of diet-induced obesity, excessive fluxes of triacylglycerol (TAG) in the adipose tissue impairs its proper storage function of the lipids which eventually leads to their ectopic accumulation in peripheral tissues, such as the liver, the muscle, and

the cardiovascular system (2, 3). The lipotoxicity in the peripheral tissues/organs ultimately leads to pro-inflammatory reaction and insulin resistance in the whole body (3). Lipid droplets (LDs) are cytosolic organelles that serve as the major energy reservoir in adipocytes as well as other cell types (4). They are formed by a core of neutral lipids surrounded by a monolayer of phospholipids (4, 5, 6, 7). The surface of the LDs is sealed by a set of enzymes and regulatory proteins (7, 8). Up to now, about 100–150 lipid droplet-associated proteins have been discovered in mammalian cells by various proteomics-based approaches (9). Importantly, the dynamics of the lipid droplets mediated by their surface proteins is tightly coupled to the whole-cell energy homeostasis (9, 10). Therefore, knowing the detailed functions of these proteins is crucial to deal with the systemic metabolic disorders.

Identification of novel players in lipid droplet dynamics may help to provide molecular insights for treatment of metabolic diseases due to abnormal storage or dissipation of lipids (4, 11). By using a mass spectrometry (MS) approach, we recently identified Ces1d, also previously annotated as triacylglycerol hydrolase (TGH) and Ces3, as one of the major lipid droplet targeted proteins in response to β-adrenergic signaling stimulation in both WAT and BAT (12). Ces1d and its human orthologue CES1 belong to the large carboxylesterase family of isoenzymes that are highly conserved among species (13, 14). They have been reported to be enriched in the liver where their metabolic functions have been well characterized (15). The three-dimensional structure of Ces1d reveals that it contains a catalytic domain surrounded by α/β regulatory domains (review by Lian et al [2018] (13)). Specifically, its large, yet flexible binding pocket in the catalytic region confers the ability of Ces1d to hydrolyze a variety of structurally distinct ester substrates. Of note, a putative neutral lipid binding domain close to its central catalytic region suggests its direct enzymatic function on neutral lipids (16, 17). In the liver, Ces1d has been shown to locate predominantly in ER where it participates in the mobilization of TG for the assembly and

---

[1]Center for Metabolic and Degenerative Diseases, The Brown Foundation Institute of Molecular Medicine for the Prevention of Human Diseases, University of Texas Health Science Center at Houston, Houston, TX, USA   [2]Center for Precision Health, School of Biomedical Informatics, University of Texas Health Science Center at Houston, Houston, TX, USA   [3]Metabolomic Core Facility, Department of Bioinformatics and Computational Biology, The University of Texas MD Anderson Cancer Center, Houston, TX, USA   [4]Department of Integrative Biology and Pharmacology, The University of Texas Health Science Center at Houston, Houston, TX, USA   [5]Program in Biochemistry and Cell Biology, MD Anderson Cancer Center-UTHealth Graduate School of Biomedical Sciences, Houston, TX, USA   [6]Group on Molecular and Cell Biology of Lipids, Department of Pediatrics, University of Alberta, Edmonton, Canada

Correspondence: kai.sun@uth.tmc.edu
Li Yang's present address is Department of Molecular and Cellular Biology, Baylor College of Medicine, Houston, TX, USA.
*Gang Li and Xin Li contributed equally to this work.

---

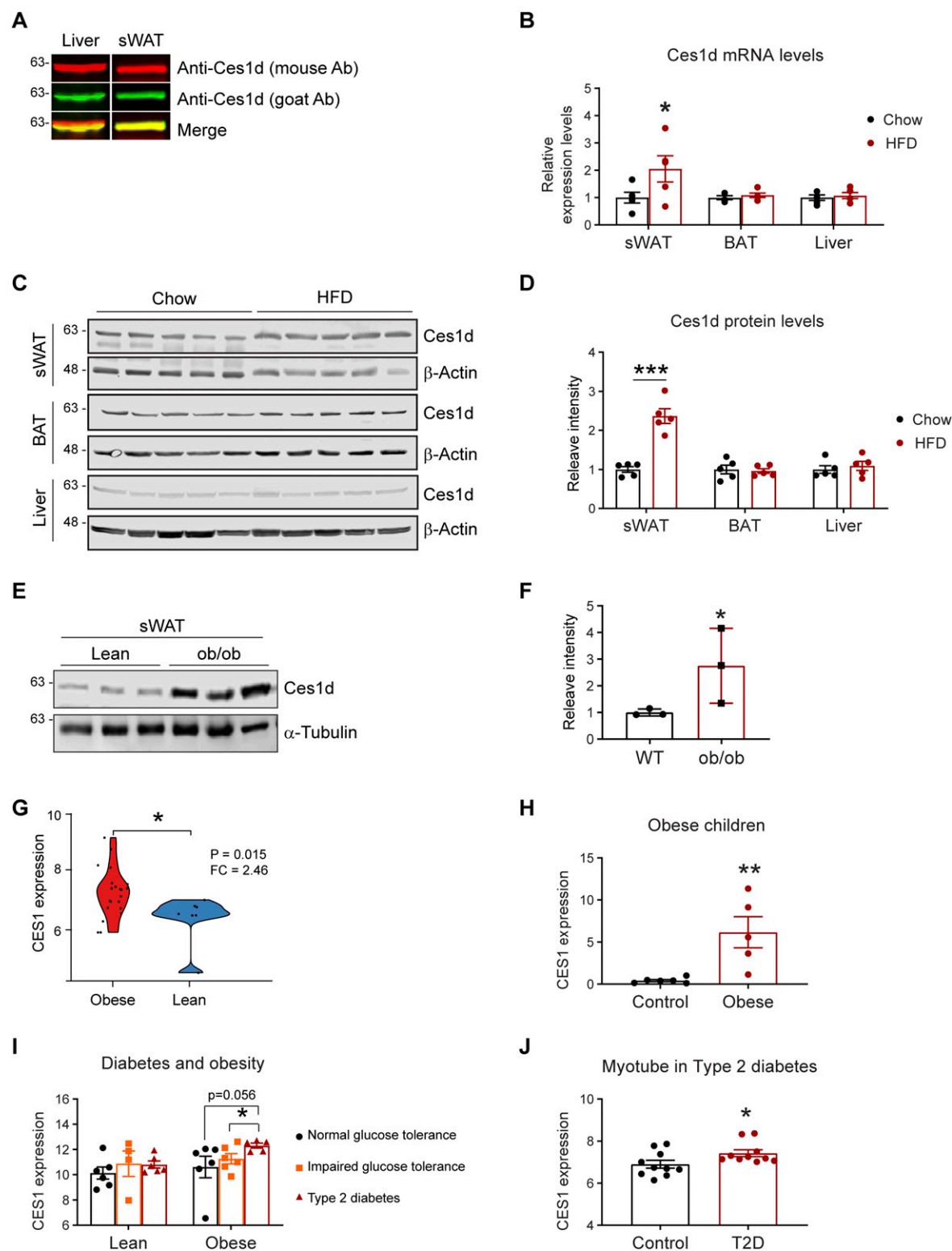

**Figure 1. The levels of Ces1d increase in adipose tissues during obesity.**

**(A)** Western blotting (WB) analysis of Ces1d in 50 μg proteins of the tissue lysates collected from the liver and subcutaneous white adipose tissue (sWAT) of the C57BL/6J wild-type (WT) mice using mouse α-Ces1d antibody (red) and goat α-Ces1d antibody (green). **(B)** Comparison of the mRNA levels of Ces1d by qPCR in the samples of sWAT, brown adipose tissue (BAT), and liver collected from WT mice fed by regular chow or high fat diet (HFD) for 14 wk (n = 5 per group, each point represents a biology replicate, representative of three repeats). Data are represented as mean ± SEM, t test, *P < 0.05. **(C)** WB analysis of Ces1d in the lysates from the sWAT from WT mice fed by regular chow or HFD for 14 wk. β-Actin was used as loading controls (n = 5 per group, representative of three repeats). **(D)** Quantification of the band intensity for Ces1d in (c) by

secretion of ApoB-containing VLDL (13, 17, 18, 19, 20, 21, 22). The effect of Ces1d on dyslipidemia and steatohepatitis has been further studied in global and liver-specific Ces1d deficient mice (13, 18, 19, 23). In line with the findings about Ces1d-mediated adverse effects on lipid and glucose metabolism in the ER, loss of function of Ces1d in liver or in the whole-body was demonstrated to improve insulin sensitivity and protect the mice from diet-induced liver steatosis and inflammation (23, 24).

Compared with its active enzymatic function in the liver, Ces1d has been considered to solely participate in basal lipolysis in the adipose tissue (25, 26). However, our recent study challenged this proposed the function of Ces1d in adipose tissue. We found that Ces1d translocates onto lipid droplets and promotes enhanced lipolysis with an efficiency that is comparable to ATGL in response to cold exposure and β-adrenergic activation in adipose tissues (12, 27). Although our previous findings provide novel insights into the beneficial effects of Ces1d in adipose tissue, the study was limited to a condition of transient β-adrenergic signaling stimulation. The potential function of Ces1d under more physio/pathologically relevant conditions, especially in response to the nutritional fluctuations in adipose tissue needs further investigation. Of note, we found that more Ces1d translocates onto lipid droplets in both adipose tissue and liver upon HFD feeding, suggesting its potential role in lipid catabolism under nutritional stress condition (12).

In this study, we revealed that there is more Ces1d existing on the lipid droplets than in the ER in adipose tissue in obese mice. Its levels are dramatically increased in the obese patients with type 2 diabetes. To study the function of lipid droplet-associated Ces1d on metabolism, we generated an adipose tissue (fat)–specific Ces1d knockout (FKO) mouse model to study the molecule in lieu of diet-induced obesity. The FKO mice exhibited larger lipid droplets, increased fat masses, as well as augmented body weight gains when challenged with HFD. Moreover, the mice presented with exacerbated liver steatosis. As a result, the FKO mice showed a metabolically unhealthy phenotype including impaired glucose and lipid metabolism, compromised mitochondrial function, higher level of inflammation and ultimately, systemic insulin resistance. Therefore, our findings reveal a key role of Ces1d in adipose tissue in protecting the mice from diet-induced obesity and hence highlight it as an attractive target to deal with obesity-related metabolic disorders.

# Results

### Ces1d (CES1) expression levels are up-regulated in obese patients with T2D

The function of Ces1d in ER has been characterized in the liver. Here, we sought to investigate its regulations in adipose tissue. Given that multiple Ces1 isoforms with different functions exist in the metabolically active issues, it is necessary to distinguish Ces1d from other isoforms with a specific antibody. For the Western blots, we tested two anti-Ces1d antibodies on the liver and adipose tissue samples collected from wild-type (WT) mice: the one is an anti-Ces1d antibody (goat, green) which specifically detects Ces1d but no other isoforms, whereas the other one (mouse, red) also recognizes other Ces1 isoforms. The properties of the antibodies are based on the testing results from the manufacture. The results revealed that the bands recognized by both antibodies completely merged as shown by the single yellow color for the samples of adipose tissue, whereas they only partially merged for the samples of the liver (Fig 1A). The result confirmed that there are some isoforms of Ces1 existing in the liver but not adipose tissue (13). We used the anti-Ces1d (goat) antibody for all the other tests throughout the study.

We next compared the levels of Ces1d in adipose tissue upon a high fat-diet (HFD) challenge for 14 wk. The results reveal that its mRNA levels were up-regulated, whereas its protein levels were significantly increased in the sWAT (Fig 1B–D) upon HFD feeding for 14 wk. Intriguingly, neither the mRNA nor the protein levels were changed in the BAT and the liver (Fig 1B–D). Ces1d levels were also dramatically increased in the sWAT of the *ob/ob* mice (Fig 1E and F). To address the clinical significance of the changes of Ces1d in obese models, we further analyzed CES1, the human homolog for Ces1d, in the patients under different physio/pathological conditions from the Gene Expression Omnibus (GEO) database (28, 29, 30, 31, 32). Based on the analyses, in the general populations, the expression levels of CES1 are dramatically higher in the obese patients when compared with lean individuals (Fig 1G). This is also true specifically in the children (Fig 1H). Interestingly, whereas there are no significant differences between the patients with normal glucose tolerance, impaired glucose tolerance, and type 2 diabetes (T2D) in the lean population, the individuals with T2D have significantly higher expression levels of CES1 in the obese population (Fig 1I). Of note, the myotube cells isolated from the T2D patients also expressed higher levels of CES1 (Fig 1J). In summary, Ces1d is up-regulated in the late phase during obesity development in both animal models and humans.

### Ces1d shows unique lipid droplet localization in adipose tissue

Ces1d has been reported to participate in basal lipolysis in the adipose tissue (25, 26). However, its lipolytic regulation in response to nutritional stress remains elusive. To determine whether it targets the lipid droplets in adipose tissue, the subcutaneous WAT (sWAT) collected from the HFD-fed mice was prepared for immunofluorescence staining (IF) with anti-Ces1d and anti-PLIN1 (lipid droplet marker protein) antibodies. The results reveal that under both HFD and chow feeding conditions, significant amount of Ces1d translocated onto or got proximate to the lipid droplets, as

ImageJ (n = 5 per group, each point represents a biology replicate). Data are represented as mean ± SEM, *t* test, ***P < 0.001. **(E)** WB analysis of Ces1d in the lysates from the sWAT from *ob/ob* mice fed by regular chow for 14 wk. α-Tubulin was used as loading controls (n = 3 per group, representative of three repeats). **(F)** Quantification of the band intensity for Ces1d in (c) by ImageJ (n = 3 per group, each point represents a biology replicate). Data are represented as mean ± SEM, *t* test, *P < 0.05. **(G)** The CES1 expression levels in the adipose tissues from morbidly obese patients or controls. The data was obtained from the Gene Expression Omnibus database (GDS3679). n = 7–21 per group. **(H)** The CES1 expression levels in adipose tissues from the obese and normal-weight prepubertal children (GDS3688). n = 5~6 per group. **(I)** The CES1 expression levels in adipose tissues from the lean or obese subjects with normal, impaired glucose tolerance, or type 2 diabetes (GDS3961). n = 4–6 per group. **(J)** The CES1 expression levels in the myotube cell lines established from type 2 diabetes (T2D) or control subjects (GDS3681). n = 10 per group.
Source data are available for this figure.

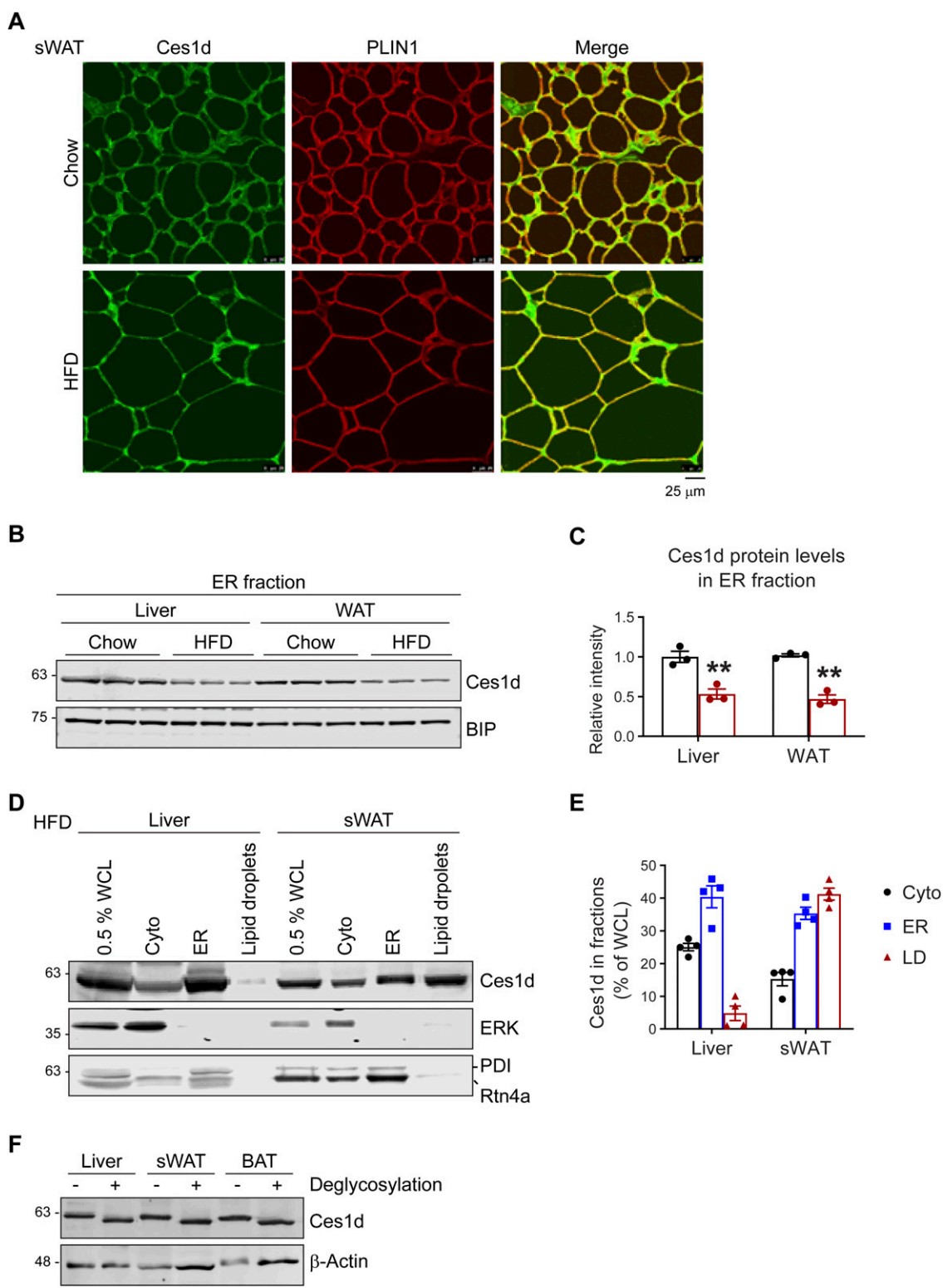

**Figure 2. Ces1d localizes on both ER and lipid droplets in adipose tissue.**
**(A)** Co-immunofluorescence (Co-IF) staining with α-Ces1d (green) and α-Perilipin-1 (PLIN1, red) (lipid droplet marker protein) antibodies on the sWAT from WT mice fed by regular chow or HFD (representative of six fields, experiments were repeated for three times). Scale bars: 25 μm. **(B)** WB analysis of Ces1d in the ER fractions isolated from the liver and WAT from WT mice fed by regular chow or HFD. BIP was used as the loading control for the ER fraction (n = 3 per group, representative of three repeats). **(B, C)** Quantification of the band intensity for Ces1d in (B) (n = 3 per group, each point represents a biology replicate). Data are represented as mean ± SEM, t test, **P < 0.01. **(D)** WB analysis of Ces1d in different cell fractions, including whole cell lysates (WCL), cytoplasm (Cyto), ER, and lipid droplets of the liver and sWAT from the wild-type

demonstrated by the colocalization of Ces1d with PLIN1 (Fig 2A). Intriguingly, Ces1d levels decreased in the ER fraction in both liver and WAT under HFD (Fig 2B and C). However, whereas its levels decreased in the ER fraction (Fig 2B and C), they showed significant increases in the whole cell fraction in WAT under HFD (Fig 1C and D). The disparity between the liver and sWAT suggested that Ces1d might translocate into other cellular compartments in adipose tissue. To test the hypothesis, we collected the samples of the ER, lipid droplets, and cytosol from the liver and adipose tissue of the HFD-fed mice and compared Ces1d levels in the fractions. The results indicated that significant amounts of Ces1d co-isolated with the lipid droplets in the sWAT, whereas most of Ces1d remained in the ER in the liver in response to the HFD challenge (Fig 2D and E). ERK was used as the loading control for the cytosolic fractions, whereas PDI and Rtn4α were used as the loading controls for the ER fractions (Fig 2D). Glycosylation modification of Ces1d might be correlated to its ER retention (13). However, analysis of the glycosylation levels collected from different tissues of the HFD fed mice with de-glycosylation enzymes did not show any differences, as demonstrated by the same pattern of the lower bands upon removal of the glycans (Fig 2F).

### Ces1d exhibits hydrolase activity on lipid droplets ex vivo

Previous study showed that Ces1d hydrolyzes TG to produce different species of fatty acids in vitro (33). Here, we aimed to study the hydrolytic function of Ces1d on the lipids isolated from the mice. The histidine-tagged Ces1d (His-Ces1d) and His-ATGL, overexpressed and purified from the Hela cells, were incubated with same amounts of lipids (200 ml) isolated from the sWAT of the WT mice. The lipids from the incubation were extracted and separated by TLC and further eluted from the plate for LC-MS/MS analysis. The results suggest that similar levels of total FFAs were produced by Ces1d as ATGL on the TLC plate (Fig 3A). The LC-MS/MS analysis revealed that the digested FFA species by Ces1d were different from ATGL (Fig 3B). Specifically, hydrolysis of lipids by Ces1d produced more short to medium-chain saturated FFAs when compared with ATGL (Fig 3B). Furthermore, products of the long-chain unsaturated FFAs showed different patterns (Fig 3B–E). Intriguingly, both enzymes exhibited high efficiency on producing the polyunsaturated FFAs, such as the C18 precursors linoleic acid (18:2n6) and α-linolenic acid (18:3n6) (Fig 3C and D), whereas ATGL produced more 22:1 than Ces1d (Fig 3E). Of note, linoleic acid was previously reported to bind to HNF4α in fed mice (34). Collectively, our results demonstrated the direct hydrolytic activity of Ces1d on WAT lipids.

### Deficiency of Ces1d in adipose tissue leads to larger fat mass and fatty liver

To address the specific role of Ces1d in adipose tissue, an adipose tissue–specific Ces1d knockout mouse model, so called the Fat-

Ces1d knockout (FKO) mouse was generated by breeding the adiponectin-Cre mice with the Ces1d[flx/flx] mice (Fig 4A) (12). The littermate Ces1d[flx/flx] mice were setup as the WT controls. Western blotting results confirmed that the endogenous Ces1d is sufficiently ablated in the adipose tissues, including the WAT and BAT, but not affected in other tissues, such as the liver (Fig 4B).

To study the metabolic role of Ces1d in adipose tissue, 8-wk-old FKO and WT mice were challenged with or without HFD for 14 wk. Notably, the FKO mice were slightly heavier even before the HFD challenge (Fig 4C). Intriguingly, the mice did not exhibit differences on energy expenditure (Fig S1J–S). The body weight differences between the groups are likely due to larger fat masses and reduced lean masses in the FKO (Fig 4D and E). The biopsies of the tissues indicated that the sizes of the sWAT and BAT were larger in the FKO mice upon HFD challenge (Fig 4F). Furthermore, the FKO mice developed more severe fatty liver (Fig 4G). Consistently, H & E staining further revealed that the lipid droplet sizes were much larger in the sWAT, eWAT, and BAT of the FKO mice (Fig 4H). More lipid droplets with larger sizes were detected in the liver of the FKO mice upon HFD challenge (Fig 4H). In summary, the FKO mice exhibited increased fat masses with larger lipid droplets when compared with the WT controls. However, the mice did not exhibit significant differences on energy expenditure. Furthermore, HFD challenge further led to exacerbated hepatic lipid deposition in the FKO mice.

### Deficiency of Ces1d in adipose tissue impairs lipid homeostasis upon HFD challenge

Our results above revealed that Ces1d is a potent hydrolase that might affect lipid homeostasis. Therefore, deficiency of Ces1d in adipose tissue may lead to dysregulations in lipid metabolism. Interestingly, most of the fasting circulating lipid components, including TG, FFA, cholesterol, HDL, and LDL/VLDL did not change dramatically under regular chow feeding condition (Fig S1A–F), suggesting a compensatory effect of other lipases. However, the fasting circulating TG levels significantly increased in the FKO mice upon HFD challenge (Fig 5A–F). Moreover, the fasting TG levels in the liver also significantly increased in the HFD-fed FKO mice (Fig 5G–I).

### Deficiency of Ces1d in adipose tissue alters the dynamics of the lipid droplets

Because Ces1d co-isolates with lipid droplets, we then asked whether deficiency of Ces1d might affect other lipid droplet-associated proteins, including the canonical lipases. qPCR analysis revealed that whereas HFD challenge up-regulated the key lipase regulators of lipid droplet turnover, such as Pnpla2, Pipe, Mgll, and Abhd5 in the sWAT, deficiency of Ces1d in the FKO mice did not further alter their expression patterns (Fig 5J). Similar results were observed in other tissues, including BAT and the liver (Fig S1G and H),

---

mice. ERK was used as the loading control for cytosolic proteins, whereas PDI and Rtn4a were used as the loading controls for the ER proteins. 0.5% of the total WCL, 1% of the Cyto, ER, and lipid droplets extracts were loaded (representative of three repeats). **(D, E)** Quantification of the band intensity for Ces1d in (D) (n = 4 per group, each point represents a biology replicate). Data are represented as mean ± SEM. **(F)** WB analysis of Ces1d in the lysates from the liver, sWAT, and BAT from WT mice. The samples were pretreated with de-glycosylation enzymes to remove both O- and N-glycans before the analysis.

**A**

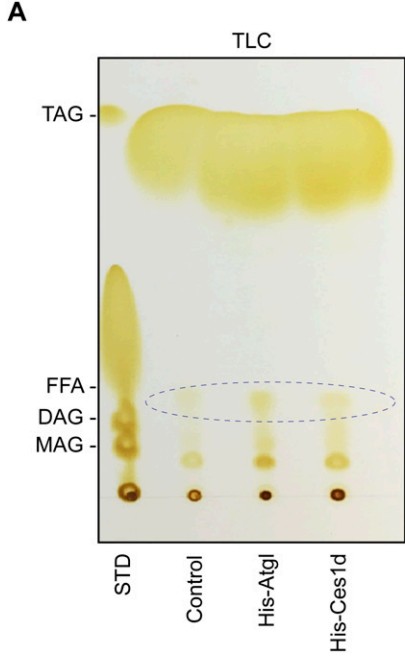

TLC

**B**

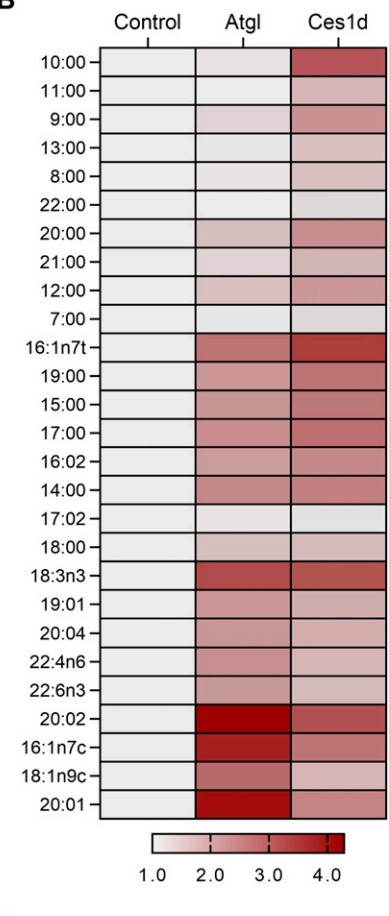

**Figure 3. Ces1d hydrolyzes lipid droplets isolated from adipose tissue of WT mice and produce unique free fatty acids (FFAs).**

**(A)** TLC of the products of the neutral lipids hydrolyzed by His-Atgl or His-Ces1d. The FFAs are circulated in the panel. The neutral lipids isolated from adipose tissue of WT mice were mixed with His-Atgl or His-Ces1d proteins that were purified from 293T cells. The mixtures were rotated at 37°C for 1 h. STD, standards of lipids; TAG, triacylglycerol; FFA, free fatty acids; DAG, diacylglycerol; MAG, monoacylglycerol. Loadings were normalized to the same quantity of biomass. **(A, B, C, D, E)** The categories of the FFAs that were identified by LC-MS/MS from the fatty acid fractions as indicated by ellipse in (A). The intensities of each fatty acid were normalized to the control group.

Source data are available for this figure.

**C**

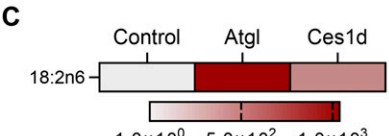

**D**

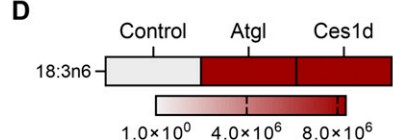

**E**

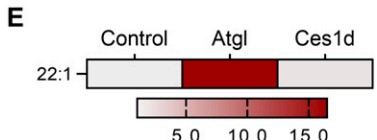

except for the levels of *Lipe* in the liver, which was significantly up-regulated in the FKO mice upon HFD challenge (Fig S1H). In line with the gene expression patterns, the protein levels of ATGL, phosphorylated-HSL, and total HSL did not show differences in the adipose tissues, including eWAT and BAT between the FKO and WT mice upon HFD challenge (Fig 5K and L).

PLINs are a family of lipid droplet–associated proteins that tightly regulate the dynamics of lipid droplets (10, 35). Intriguingly, they exhibited different regulatory patterns in the adipose tissues and the liver of the FKO mice: in the sWAT and liver, the nascent lipid droplet-associated proteins PLIN2 and PLIN3 were decreased, whereas the PLIN1 and PLIN5 were not changed (Fig 6A–D). Whereas in the BAT, PLIN1 was dramatically decreased, the PLIN2, PLIN3, and

PLIN5 were not changed in the FKO mice upon HFD challenge (Fig 6E and F).

De novo lipogenesis may lead to the enlargement of the lipid droplets. qPCR analysis revealed that the key enzymes that are involved in de novo lipogenesis, such as *Acaca*, *Fasn*, and *Scd1* in sWAT and liver did not change in the FKO mice under both chow and HFD feeding conditions (Figs 6G and S1I). However, phosphorylated ACC1 levels as well as the ratio of phospho-ACC1 and total ACC1 were dramatically increased in BAT under HFD challenge (Fig 6H and I). Consistently, the phosphorylated AMPK levels and the ratio of phospho-AMPK and total AMPK were increased (Fig 6H and J). Similar differences were also observed in the sWAT upon HFD challenge but not under chow feeding condition between the WT

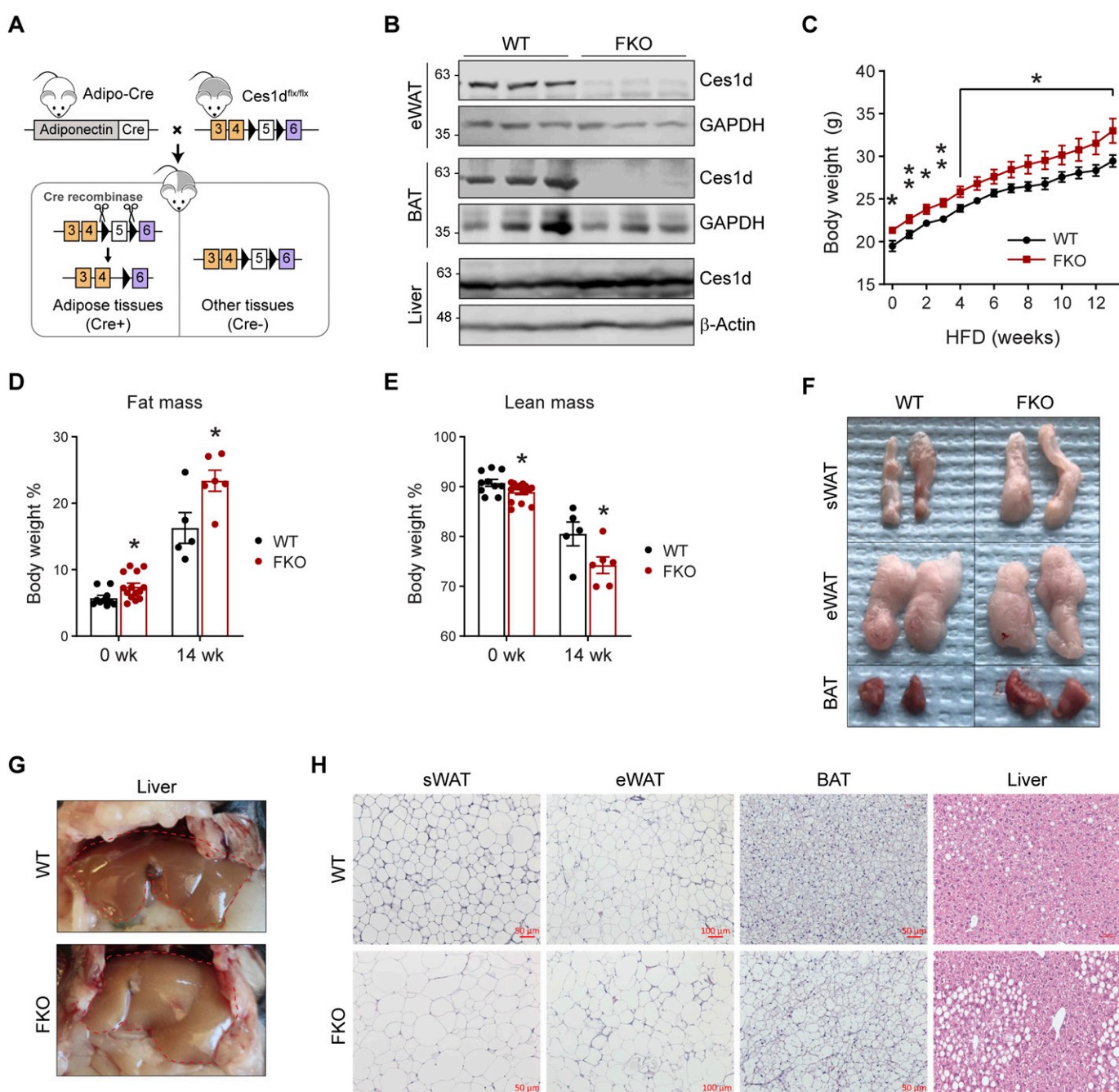

**Figure 4. Adipose tissue–specific Ces1d knockout (FKO) mice gain more body weights with larger fat masses and fatty liver.**
**(A)** Schematic representation of adipocyte-specific *Ces1d* knockout (*Adipo-Cre;Ces1d*$^{flx/flx}$) mouse model. **(B)** WB analysis for Ces1d in the lysates from epididymal WAT (eWAT), BAT, and liver of the FKO and their littermate control WT mice. GAPDH was used as the loading control for eWAT and BAT, whereas $\beta$-actin for liver. (n = 3 per group, representative of three repeats). **(C)** Body weights of the WT and FKO mice during a 14-wk HFD feeding (n = 5 in WT group and n = 10 in FKO group, each point represents a biology replicate, representative of three repeats). Data are represented as mean ± SEM, *t* test, \*$P < 0.05$, \*\*$P < 0.01$. **(D, E)** Fat mass (D) and lean mass (E) of the WT and FKO mice with or without HFD feeding (n = 5–14, each point represents a biology replicate, representative of three repeats). Data are represented as mean ± SEM, *t* test, \*$P < 0.05$. **(F)** Images of biopsies of the sWAT, eWAT, and BAT from WT and FKO mice after 14-wk HFD feeding (representative of five mice per group). **(G)** Images of biopsies of the liver from WT and FKO mice after 14-wk HFD feeding (representative of five mice per group). **(H)** H & E staining for the sWAT, eWAT, BAT, and liver from WT and FKO mice after 14-wk HFD feeding (representative of five fields of the samples from five mice per group). Scale bars: 50, 100 $\mu$m in eWAT.
Source data are available for this figure.

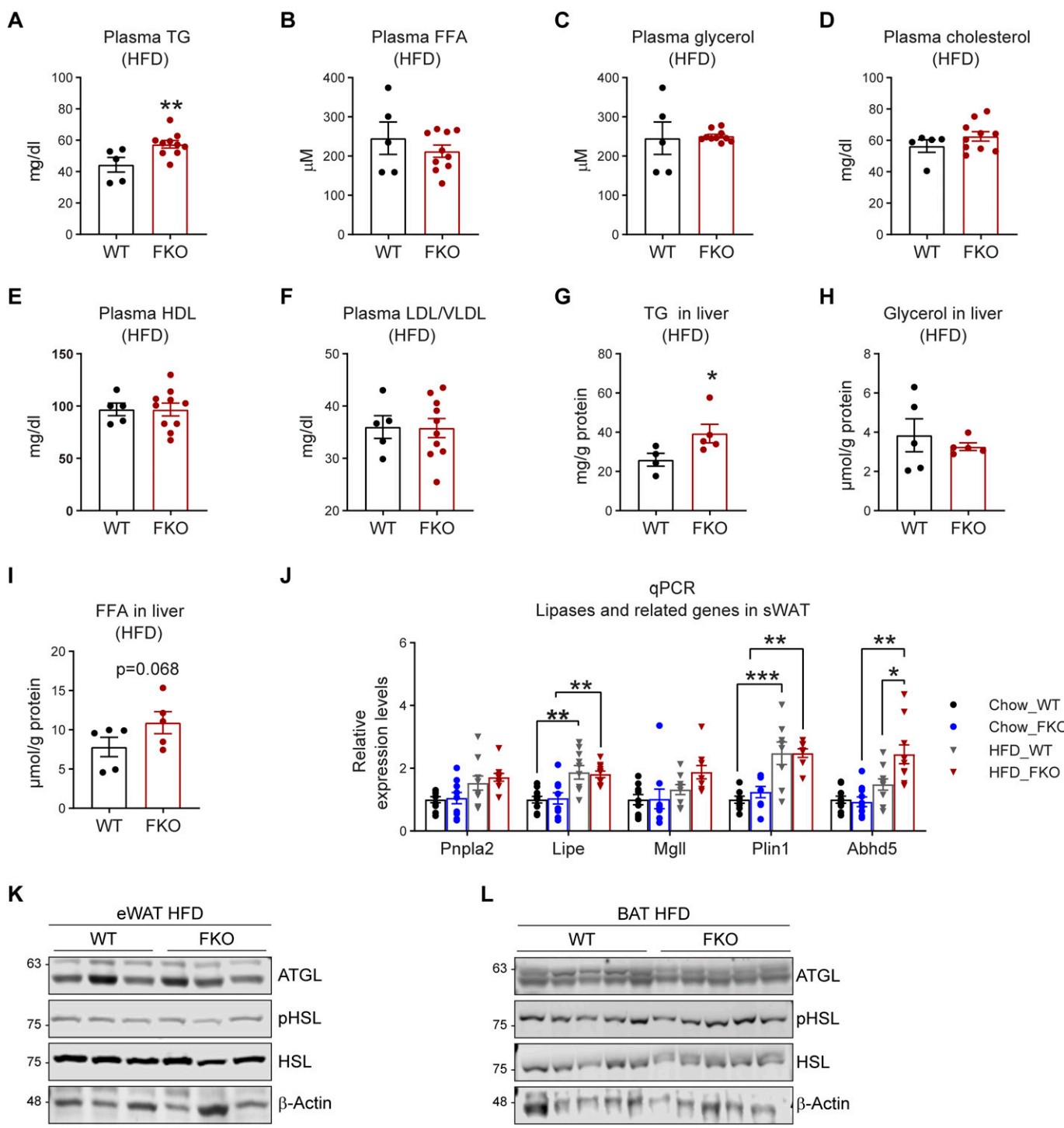

**Figure 5. Deficiency of Ces1d in adipose tissue impairs lipid homeostasis.**
**(A, B, C, D, E, F)** The levels of triglyceride (TG) (A), free fatty acid (FFA) (B), glycerol (C), cholesterol (D), high-density lipoproteins (HDL) (E), and low-density lipoproteins and very low-density lipoproteins (LDL/VLDL) (F) in the plasma of WT and FKO mice after 14-wk HFD feeding (n = 5 in WT group and n = 10 in FKO group, each point represents a biology replicate, representative of three repeats). Data are represented as mean ± SEM, *t* test, **$P < 0.01$. **(G, H, I)** The levels of TG (G), FFA (H), and glycerol (I) in the liver of WT and FKO mice after 14-wk HFD feeding (n = 5 in WT group and n = 4 in FKO group, each point represents a biology replicate). Data are represented as mean ± SEM, *t* test, *$P < 0.05$. **(J)** qPCR analysis for the mRNA levels of lipases and related genes including *Pnpla2*, *Lipe*, *Mgll*, *Plin1*, and *Abhd5* in the sWAT of WT and FKO mice fed on regular chow or HFD (n = 5 per group, each point represents a biology replicate, representative of three repeats). Data are represented as mean ± SEM, *one-way ANOVA*, *$P < 0.05$, **$P < 0.01$, ***$P < 0.001$. **(K)** WB analysis of ATGL, pHSL, and HSL in the lysates from the eWAT of WT and FKO mice fed by HFD. β-Actin was used as loading control (n = 3 per group, representative of three repeats). **(L)** WB analysis of ATGL, pHSL, and HSL in the lysates from the BAT of WT and FKO mice fed by HFD. β-Actin was used as loading control (n = 5 per group, representative of three repeats).
Source data are available for this figure.

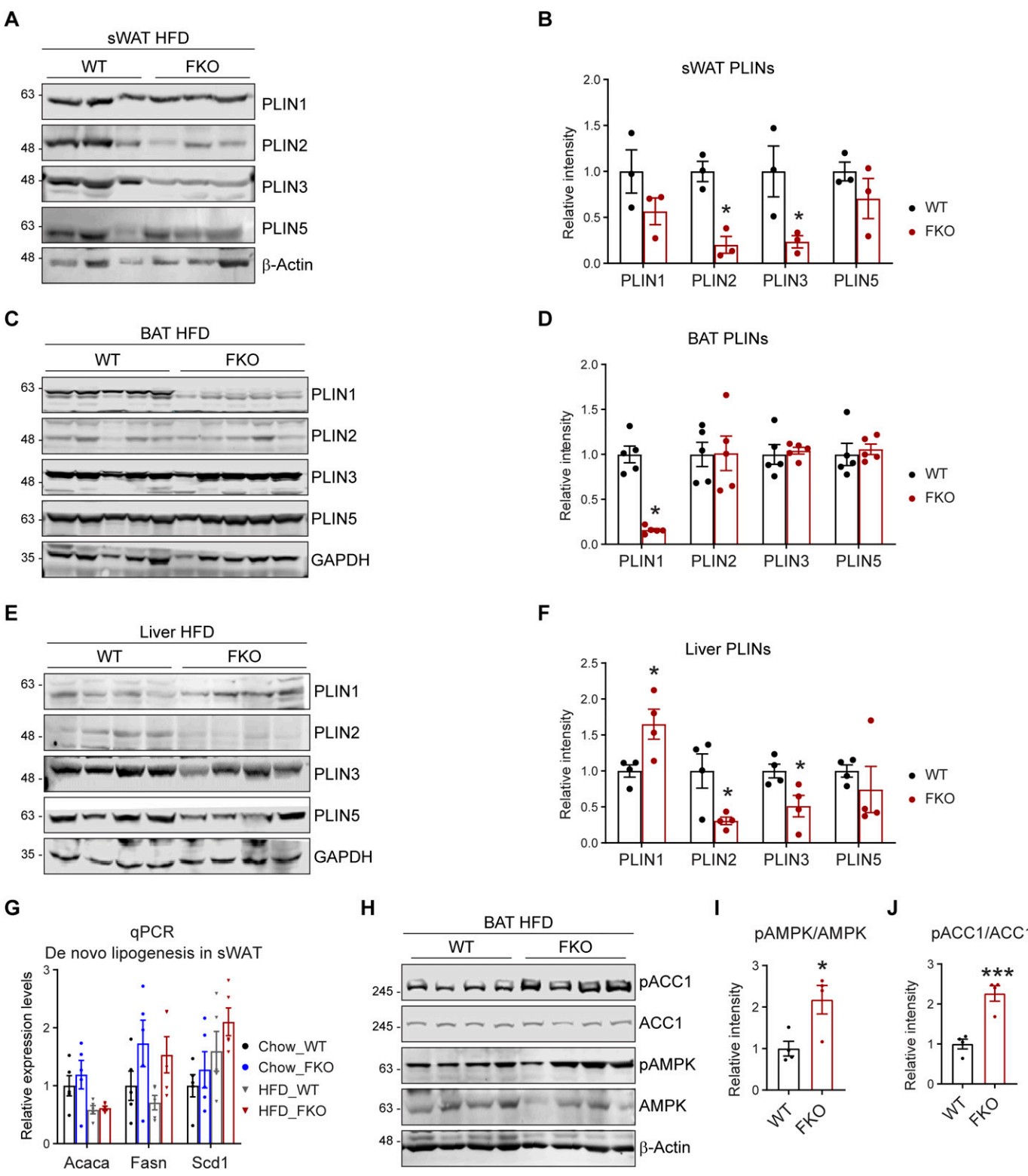

**Figure 6. Deficiency of Ces1d in adipose tissue affects the dynamics of lipid droplets.**
**(A)** WB analysis of Perilipins (PLINs) including PLIN1, PLIN2, PLIN3, and PLIN5 in the lysates from the sWAT of WT and FKO mice fed on HFD. β-Actin was used as loading control (n = 3 per group; representative of three repeats). **(A, B)** Quantification of the band intensity in (A) (n = 3 per group, each point represents a biology replicate). Data are represented as mean ± SEM, *t* test, *P < 0.05. **(C)** WB analysis of PLIN1, PLIN2, PLIN3, and PLIN5 in the lysates from the liver of WT and FKO mice fed on HFD. GAPDH was used as loading control (n = 4 per group, representative of three repeats). **(C, D)** Quantification of the band intensity in (C) (n = 4 per group, each point represents a biology replicate). Data are represented as mean ± SEM, *t* test, *P < 0.05. **(E)** WB analysis of PLIN1, PLIN2, PLIN3, and PLIN5 in the lysates from the BAT of WT and FKO mice fed

and FKO mice (Fig S2A–D). However, we did not find changes in the liver of the FKO mice (Fig S2E–H). In summary, we found that some key factors that regulate the biogenesis and dynamics of lipid droplets were changed in the FKO mice.

## Deficiency of Ces1d in adipose tissue affects the systemic lipid metabolic pathways

To further address the key role of adipose tissue–specific Ces1d in systemic lipid metabolism, a lipidomic analysis was performed on the sera collected from FKO and WT mice upon HFD challenge. In total, about 944 lipid species were analyzed and compared (Fig 7A). Among them, 11 lipid species showed significant changes (Fig 7B). These changes reflect the unique function of adipose tissue–derived Ces1d on regulation of the systemic lipid homeostasis. Of note, the total circulating TG levels were significantly increased (Fig 7B, pointed by the arrow). Interestingly, the total phosphatidic acid (PA) levels were decreased in circulation in the FKO mice, suggesting that hydrolysis of lipids by Ces1d may lead to generation of more PA species (Fig 7B, pointed by the arrow).

In addition to direct hydrolytic function on the lipids, Ces1d may also regulate lipid metabolism via other pathways. To support the notion, the pathway analysis on the lipidomic data revealed that phospholipid PA synthesis from PC was inhibited, whereas PG synthesis from PA was bolstered. Interestingly, the route of PA to PS was promoted upon ablation of Ces1d in adipose tissue (Fig 7C, refer to Table S1 for the full names of the abbreviations). As a result, the total PA levels decreased, whereas the levels of PG and CL increased in circulation (Fig 7B). Other important pathways include the decreased Cer production from SM and dhCer, the increased SM production from Cer, as well as the increased PIP and decreased PI levels that were regulated mutually (Fig 7D and E).

## Deficiency of Ces1d in adipose tissue leads to whole-body glucose intolerance and insulin resistance

Lipid dysregulation links directly to impaired glucose metabolism and insulin resistance (36). To determine the effect of adipose tissue–specific Ces1d deficiency in glucose metabolism, glucose tolerance tests (GTTs) and insulin tolerance tests (ITTs) were performed on the FKO and WT mice. The results show that even under regular chow feeding condition, fasting circulating glucose levels were significantly increased, whereas insulin levels remained normal in the FKO mice (Fig 8A and B). Furthermore, the FKO mice exhibited impaired glucose tolerance and insulin resistance during the GTT and ITT (Fig 8C–F). The FKO mice not only showed increased circulating glucose levels, impaired glucose tolerance, and insulin resistance, but also increased circulating insulin levels under HFD challenge (Fig 8G–L), suggesting a worsened condition in response to the nutritional stress. Furthermore, the

phospho-AKT levels in response to insulin injection were dramatically decreased in the liver and the muscles of the FKO mice under HFD challenge (Fig 8M–P), demonstrating impaired insulin signaling pathways in the metabolically active tissues of the FKO mice. In summary, deficiency of Ces1d in adipose tissue leads to impaired glucose metabolism as well as insulin resistance and HFD challenge worsened the conditions.

## Deficiency of Ces1d causes abnormal epigenetic regulations of metabolic pathways

Fatty acids produced by lipases not only function as substrates for energy expenditure, but also serve as important signaling molecules for lipid-sensing transcriptional factors (12, 37). Interestingly, we found that Ces1d hydrolyses lipids and produces C18 linoleic acids which had been reported to bind and activate HNF4α transcriptional activity (34). HNF4α is a master regulator for liver metabolism (34, 38, 39). We found that even though its mRNA and protein levels were not changed, its target genes, such as G6pc, Pek1, Apoc3, and Cyp7a1 were down-regulated in the liver of the FKO mice after HFD (Fig 9A and B). We also observed the similar effect in the mice fed on chow diet (Fig S3A). To further address the key role of HNF4α in Ces1d-mediated glucose and lipid metabolism, we treated the primary hepatocytes isolated from the WT or liver-specific Hnf4α knock-out mice with the serum collected from the HFD fed WT or FKO mice. The results revealed that the HNF4α target genes, including Pek1, Apoc3, and Cyp7a1 were dramatically down-regulated in the cells treated by the serum collected from FKO mice when compared with from WT mice. However, the differences were abolished in the HNF4α knock-out hepatocytes (Fig 9C). To study the direct regulation of HNF4α transcriptional activity upon Ces1d ablation, the HepG2 cells (a human liver cancer cell line) were treated with serum collected from the WT or FKO mice upon HFD challenge. A luciferase reporter assay revealed that the HNF4α specific ApoB-luciferase activity was decreased in HepG2 cells upon treatment by the serum collected from FKO mice (Fig 9D), suggesting a direct regulation of HNF4α transcriptional activity by Ces1d.

We then tested the levels of other critical transcriptional factors. qPCR results indicated that the expression levels of Srebf1, Srebf2, Mlxiplα, and Nr1h3α were not changed in the liver and adipose tissue in the FKO mice under both chow diet feeding and HFD challenge conditions (Figs 9E and S3B). Of note, even though we did not detect changes on expression levels for cAMP Response Element-Binding Protein (Creb) in the FKO mice (Fig 9F), we found that its phosphorylation levels as well as the ratio of phosphorylated and total CREB were significantly decreased in sWAT, BAT, and the liver of the FKO mice (Fig 9G–L).

Bearing the changes of the key transcriptional regulators in mind, we performed an unbiased screen for the regulated genes in the WAT of the FKO mice upon HFD challenge by using the RNA-seq

---

on HFD. GAPDH was used as loading control. (n = 5 per group, representative of three repeats). **(E, F)** Quantification of the band intensity in (E) (n = 5 per group, each point represents a biology replicate). Data are represented as mean ± SEM, *t* test, *P < 0.05. **(G)** qPCR analysis for the mRNA levels of de novo lipogenesis related genes including *Acaca*, *Fasn*, and *Scd1* in the sWAT from WT and FKO mice fed on regular chow or HFD (n = 5 in each group, each point represents a biology replicate, representative of three repeats). Data are represented as mean ± SEM, *one-way ANOVA*. **(H)** WB analysis of pACC1, ACC1, pAMPK, and AMPK in the lysates from the BAT of WT and FKO mice fed on HFD. *β*-Actin was used as the loading control (n = 4 per groups, representative of three repeats). **(H, I, J)** Quantification of the band intensity for pACC1/ACC1 ratio (I) and pAMPK/AMPK ratio (J) in (H) by ImageJ software (n = 4 per group, each point represents a biology replicate). Data are represented as mean ± SEM, *t* test, *P < 0.05, ***P < 0.001.

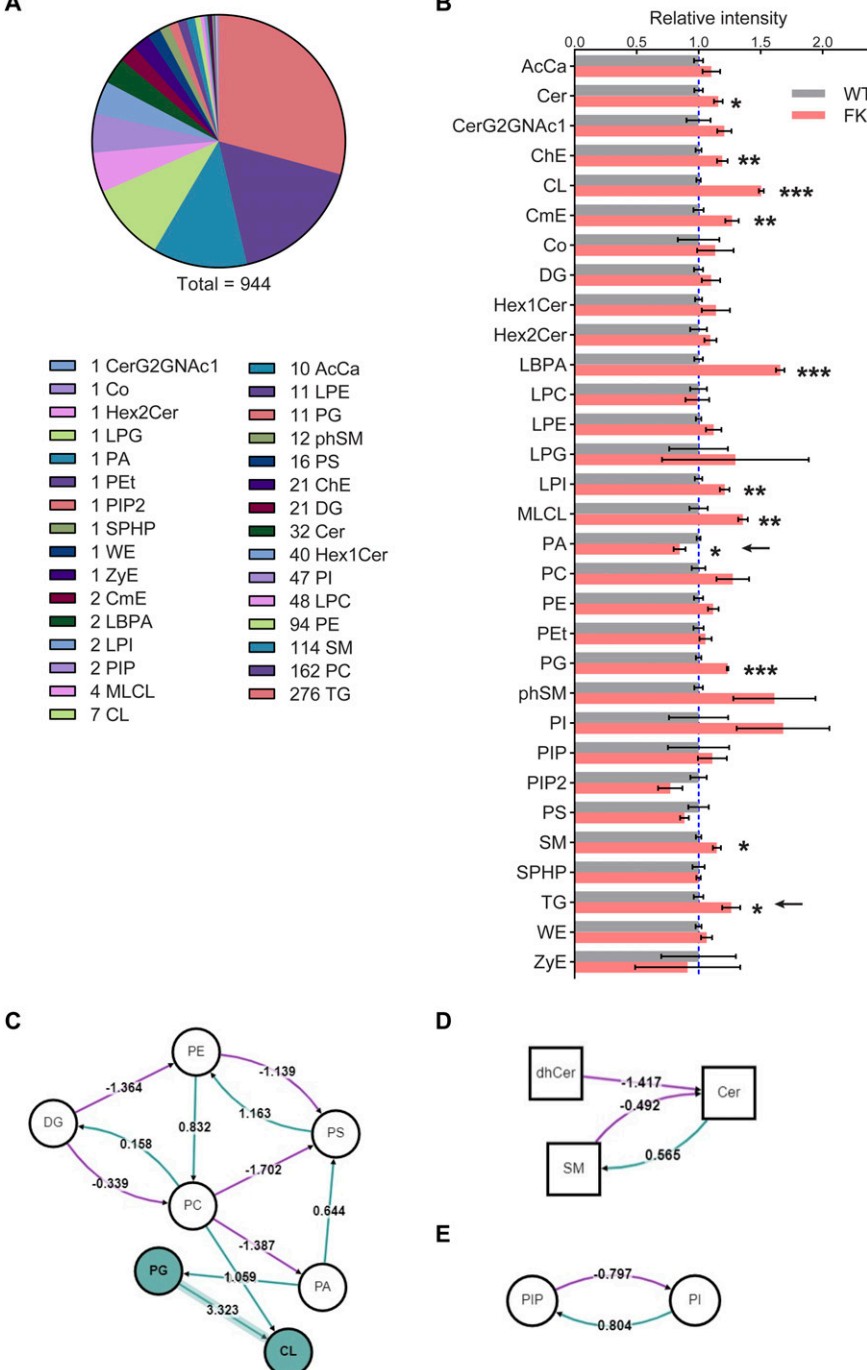

**Figure 7. Lipid profiling in the serum of FKO and WT mice upon HFD challenge by lipidomics.**
**(A)** Total 944 different lipids were detected by the lipidomic analysis in the serum from WT and FKO mice fed on HFD for 14 wk (n = 3 per group; for each sample, sera from two or three mice were pooled). **(B)** The relative intensities of the lipid species for the lipidomic analysis in the serum from WT and FKO mice fed on HFD (n = 3 per group; for each sample, sera from two or three mice were pooled). Data are represented as mean ± SEM, t test *P < 0.05; **P < 0.01; ***P < 0.001. **(C, D, E)** The lipid metabolic pathway analysis using all the detected molecules from lipidomics. Node shape: circle, glycerolipids and glycerophospholipids (including DG, PE, PS, PC, PA, PG, CL, PIP, and PI); square, sphingolipids (including dhCer, Cer, and SM). Edge color: pink means negative regulation, whereas green means positive regulation. Source data are available for this figure.

approach. The results revealed total 559 genes were down-regulated and 4,111 genes were up-regulated in response to the deficiency of Ces1d in the adipose tissue (Figs 10A and S3C). The analysis for the down-regulated genes revealed that the impaired pathways include mitochondrial oxidative phosphorylation, respiratory chain, and formation of the mitochondrial complexes/matrix etc. (Fig 8B and C and Table S2). All of the changes reflected a consequence of impaired lipid and glucose metabolism on mitochondrial functions.

## Deficiency of Ces1d causes pathological changes in adipose tissue and liver

Excessive lipid burden may cause pathological changes that shape an unhealthy microenvironment in adipose tissue (2). To determine the pathological consequences upon Ces1d deficiency in adipose tissue, the changes of adipogenesis, fibrosis, and pro-inflammatory factors were detected by qPCR. The results revealed that *Pdgfa*, *Pdgfb*, and *Zfp423* were significantly up-regulated in the sWAT,

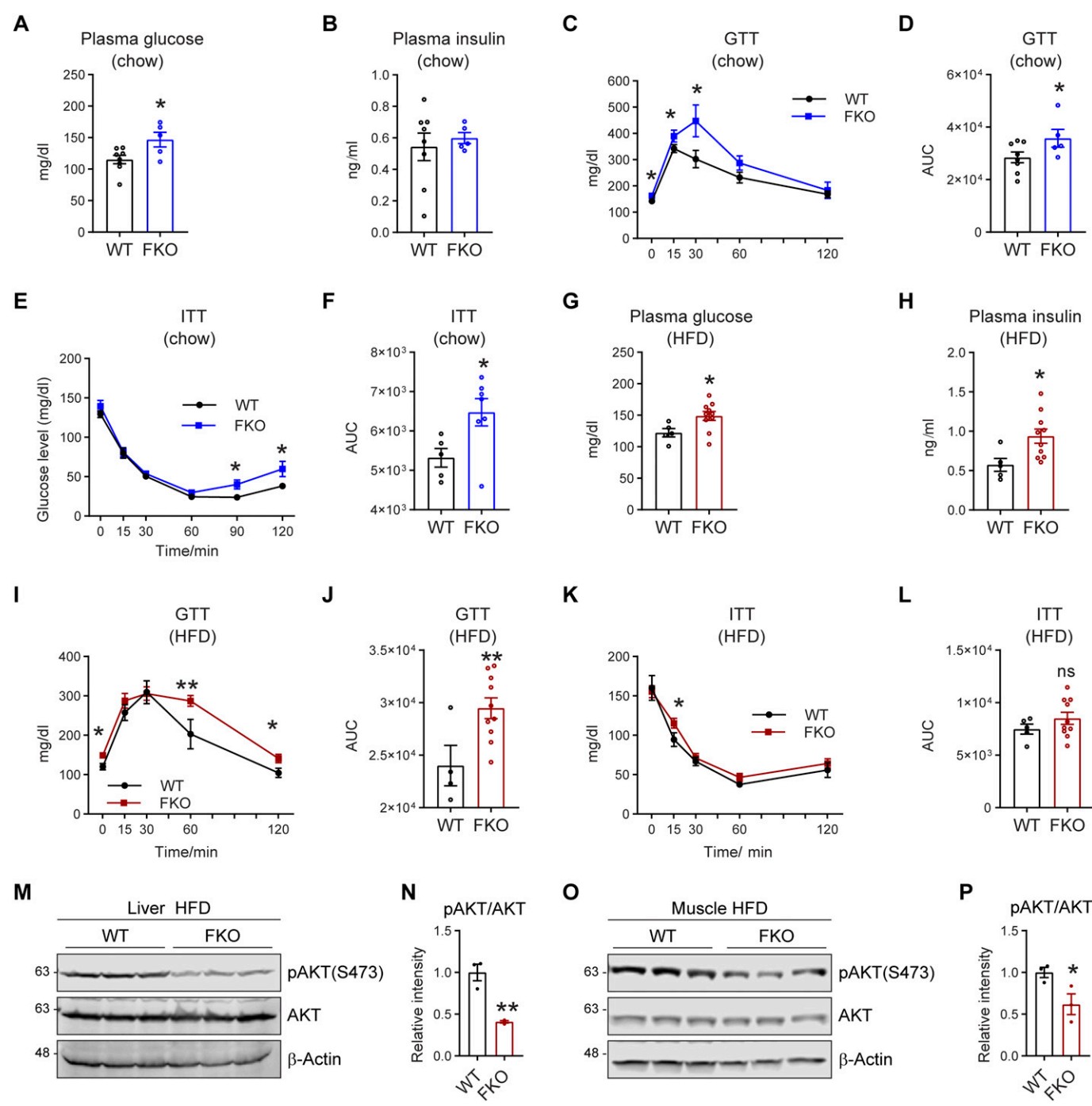

**Figure 8. FKO mice exhibit impaired glucose tolerance and insulin sensitivity.**
**(A)** Fasting glucose levels in the plasma of WT and FKO mice fed on regular chow (n = 8 in WT group and n = 5 in FKO group, each point represents a biology replicate, representative of three repeats). Data are represented as mean ± SEM, *t* test, *P < 0.05. **(B)** Fasting insulin levels in the plasma of WT and FKO mice fed on regular chow (n = 8 in WT group and n = 5 in FKO group, each point represents a biology replicate, representative of three repeats). Data are represented as mean ± SEM, *t* test. **(C)** Blood glucose levels during the glucose tolerance test in WT and FKO mice fed on regular chow (n = 8 in WT group and n = 5 in FKO group). Data are represented as mean ± SEM. *t* test, *P < 0.05. **(C, D)** The area under the curve (AUC) from (C) (n = 8 in WT group and n = 5 in FKO group, each point represents a biology replicate). Data are represented as mean ± SEM, *t* test, *P < 0.05. **(E)** Blood glucose levels during the insulin tolerance test in WT and FKO mice fed on regular chow (n = 8 in WT group and n = 5 in FKO group, representative of three repeats). Data are represented as mean ± SEM, *t* test, *P < 0.05. **(F)** The AUC of (E) (n = 8 in WT group and n = 5 in FKO group, each point represents a biology replicate). Data are represented as mean ± SEM, *t* test, *P < 0.05. **(G)** Fasting glucose levels in the plasma of WT and FKO mice fed on HFD (n = 5 in WT group and n = 10 in FKO group, each point represents a biology replicate, representative of three repeats). Data are represented as mean ± SEM, *t* test, *P < 0.05. **(H)** Fasting insulin levels in the plasma of WT and FKO mice fed on HFD (n = 5 in WT group and n = 10 in FKO group, each point represents a biology replicate, representative of three repeats). Data are represented as mean ± SEM, *t* test, *P < 0.05. **(I)** Blood glucose levels during the glucose tolerance test in the WT and FKO mice fed on HFD (n = 5 in WT group and n = 10 in FKO group, representative of three repeats). Data are represented as mean ± SEM, *t* test, *P < 0.05, **P < 0.01. **(I, J)** The AUC of (I) (n = 5 in WT group and n = 10 in FKO group,

suggesting more hyperplasia in adipose tissue of the FKO mice upon HFD feeding (Fig 11A). Furthermore, the pro-fibrotic genes, such as *Col3a1*, *Col6a3*, and *Lox*, as well as the pro-inflammatory gene *Adgre1* were significantly up-regulated (Fig 11B and C), suggesting higher levels of local fibrosis and inflammation in the WAT of the FKO mice upon HFD challenge. In line with the results, IF showed more positive Mac-2 staining, suggesting macrophage infiltration/accumulation in the eWAT of the FKO mice (Fig 11D). Importantly, the infiltrated macrophages were in M1 pro-inflammatory phase, as verified by the more CD68 (M1 marker) positive and the less CD206 (M2 marker) positive signals in the eWAT of the FKO mice (Fig 11E). Of note, the circulating leptin, but not the adiponectin, levels were significantly increased in the FKO mice upon HFD challenge (Fig S4A and B), suggesting the impaired secretory function of the adipose tissue in response to lack of Ces1d. The similar abnormalities of the pro-fibrotic and pro-inflammatory genes were also detected in the liver (Fig 11F and G). Indeed, more macrophages were infiltrated in the liver, as indicated by the Mac-2 staining in the liver of FKO mice upon HFD challenge (Fig 11H). Collectively, Ces1d deficiency in adipose tissue leads to pathological changes, including fibrosis and inflammation in the metabolically active tissues.

Taken together, a working model is proposed based on the results. In the normal adipose tissue, Ces1d hydrolyzes the lipid droplet TG. The produced FFAs circulate into BAT and the liver where they serve as the substrates for energy generation. Moreover, the FFAs also function as signaling molecules to activate the lipid-sensing transcriptional factors to regulate the glucose and lipid metabolic homeostasis (Fig 11I, left). In the Ces1d-deficient adipose tissue, insufficiency of lipolysis leads to the formation of larger lipid droplets in adipocytes. The excessive nutritional stress, especially under HFD challenging condition may cause increased ectopic deposition of TG in peripheral tissues. This leads to lipotoxicity, which further induces fatty liver and whitening of the BAT. Concomitantly, absence of the key signaling FFAs might blunt the proper regulations of key metabolic genes, which further worsens the metabolic adverse effects. Ultimately, the mice develop whole-body insulin resistance (Fig 11I, right).

## Discussion

Lipolysis in adipose tissue is critical for the whole-body energy homeostasis (40, 41, 42). The process exerts a major influence on the metabolically healthy conditions of adipose tissue. The understanding of the regulations of TG lipolysis by canonical ATGL/

HSL/MGL axis has been deeply expanded in the past two decades (40, 43, 44). Detailed investigations revealed unexpected complexity related to the complete lipolytic pathways and their regulatory mechanism (45). Of note, recent studies have highlighted the importance of additional factors in the lipolytic process. One of these factors is Ces1d, a function of which in adipose tissue has been recently drawn significant attention (12, 27). Ces1d was initially reported to be responsible for part of non-HSL lipase activity in adipocytes in vitro whereas it was considered to solely exert basal lipolysis in the adipose tissue (25, 26, 46). Recent studies on this hydrolase have been focused on its key roles in liver metabolism in the past years (18, 19, 23, 24). In the present study, we found that Ces1d (CES1 in humans) levels are significantly increased in the type-2 diabetic patients. In line with the clinical observation, its protein levels increase during the development of obesity and in the *ob/ob* adipose tissue. We reveal that Ces1d exhibits unique cellular localization by targeting on or getting close to the lipid droplets in the adipose tissue. By using an adipose tissue–specific Ces1d knockout model (FKO), we further demonstrate that Ces1d in adipose tissue plays a major role in energy homeostasis. This is particularly true when the mice were challenged by HFD. Indeed, FKO mice gained more body weights with larger fat masses upon HFD feeding. However, the mice did not exhibit significant differences on energy expenditure. The mice exhibited lipotoxicity in the peripheral organs/tissues. Specifically, they had exacerbated liver steatosis. Furthermore, the mice exhibited metabolic inflexibility, as demonstrated by systemic glucose and lipid metabolic dysregulations. Ultimately, the mice developed whole-body insulin resistance. Therefore, our findings unveiled a previously underappreciated metabolic function of Ces1d in adipose tissue.

Ces1d has been extensively studied in the liver in the rodent models. Global or liver-specific Ces1d inactivation decreases circulating lipids without inducing severe lipotoxicity in other tissues (18, 47). Specifically, elimination of Ces1d in hepatocytes dramatically decreased levels of VLDL TG and VLDL cholesterol in circulation (18). As a result, the mice were protected against diet-induced liver steatosis and exhibited improved insulin sensitivity and enhanced energy expenditure (24, 47). Moreover, ablation of Ces1d attenuated nonalcoholic steatohepatitis (NASH) caused by deletion of *Pemt* gene and in *Ldlr⁻/⁻* mice (23). Apparently, deficiency of Ces1d in adipose tissue leads to different metabolic phenotype: the mice exhibited impaired lipid and glucose metabolism and consequently, the mice developed obesity and related metabolic disorders. The results suggest dual roles of Ces1d in the two metabolically active organs: adipose tissue and liver. We believe that different localization of the protein in different cells

each point represents a biology replicate). Data are represented as mean ± SEM, *t* test, **$P < 0.01$. **(K)** Blood glucose levels during the insulin tolerance test in WT and FKO mice fed on HFD (n = 5 in WT group and n = 10 in FKO group, representative of three repeats). Data are represented as mean ± SEM, *t* test, *$P < 0.05$. **(K, L)** The AUC of (K) (n = 5 in WT group and n = 10 in FKO group, each point represents a biology replicate). Data are represented as mean ± SEM, *t* test. **(M)** WB analysis of pAKT(S473) and AKT in the lysates from the liver of WT and FKO mice fed on HFD. The samples were collected 15 min later after insulin injection via i.p. *β*-Actin was used as the loading control (n = 3 per group, representative of three repeats). **(N)** Quantification of the band intensity of pAKT/AKT ratio in (M) (n = 3 per group, each point represents a biology replicate). Data are represented as mean ± SEM, *t* test, **$P < 0.01$. **(O)** WB analysis of pAKT(S473) and AKT in the lysates from the muscle of WT and FKO mice fed on HFD. The samples were collected 15 min later after insulin injection via i.p. *β*-Actin was used as the loading control (n = 3 per group, representative of three repeats). **(O, P)** Quantification of the band intensity of pAKT/AKT ratio in (O) (n = 3 per group, each point represents a biology replicate). Data are represented as mean ± SEM, *t* test, *$P < 0.05$.
Source data are available for this figure.

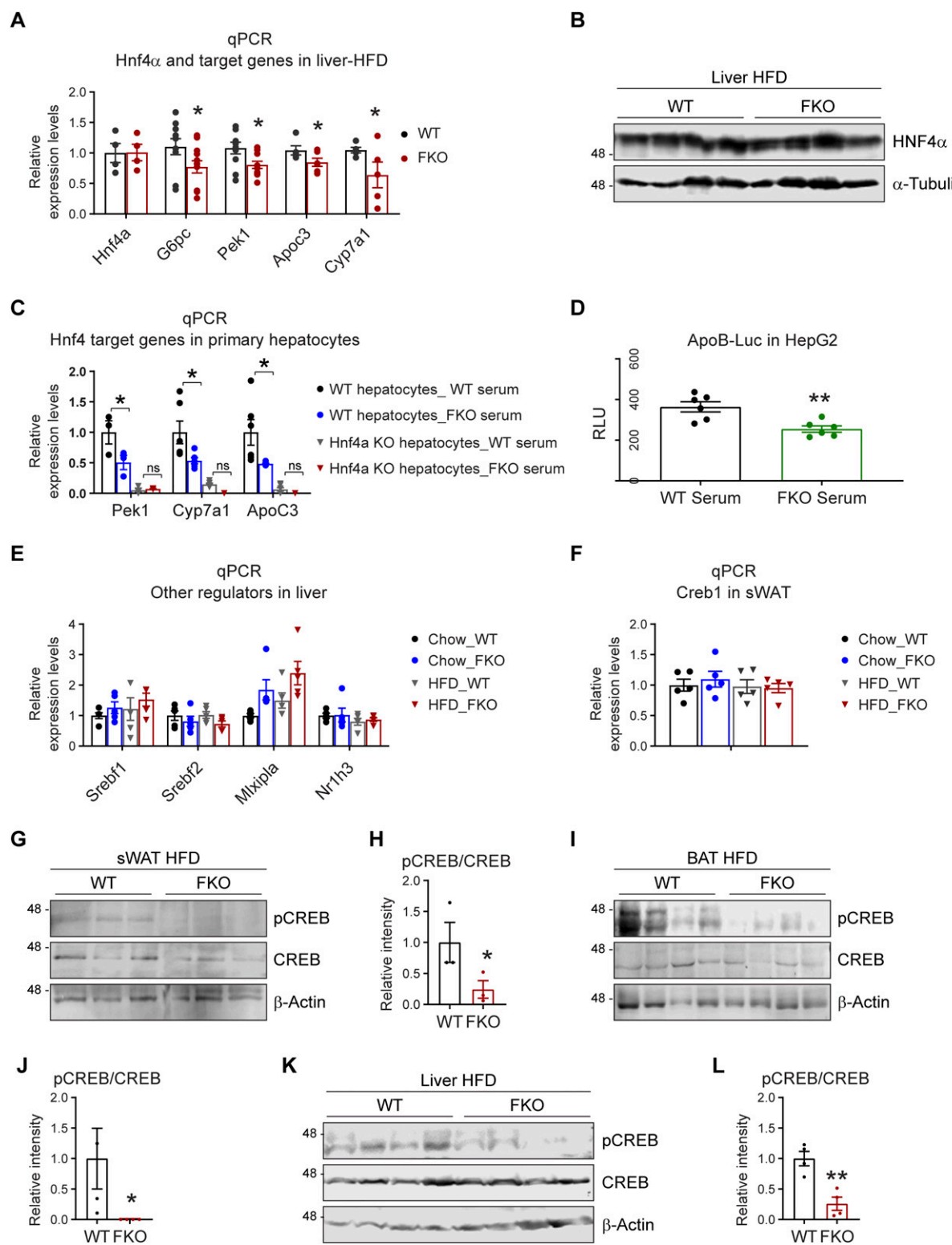

**Figure 9. Deficiency of Ces1d in adipose tissue leads to epigenetic regulations of the metabolic genes.**
**(A)** qPCR analysis for the mRNA levels of *Hnf4α* and its target genes including *G6pc*, *Pck1*, *Apoc3*, and *Cyp7a1* in the liver of WT and FKO mice fed on HFD (n = 5–10 per group, each point represents a biology replicate, representative of three repeats). Data are represented as mean ± SEM, *one-way ANOVA*, *P < 0.05. **(B)** WB analysis of HNF4α in the lysates from the liver of WT and FKO mice fed on HFD. Tubulin-α was used as loading control (n = 4 per group, representative of three repeats). **(C)** qPCR analysis for the mRNA levels of *Hnf4α* target genes including *Pck1*, *Cyp7a1*, and *Apoc3* in the primary hepatocytes of WT or liver-specific Hnf4 knockout mice treated by the serum from WT or FKO mice (n = 3–6 per group, each point represents a biology replicate, representative of three repeats). Data are represented as mean ± SEM, *t* test,

would explain the tissue specific functions of Ces1d. Ces1d predominantly localizes on ER in the liver (47, 48). The ER-localized Ces1d hydrolyses TG and provides substrates for the assembly of ApoB-containing lipoproteins, especially the VLDL in the liver (17, 33). Indeed, several studies support the active role of Ces1d in VLDL assembly and secretion from the ER (20, 21, 48). Given that the major function of hepatic VLDL is to deliver TG to peripheral tissues, its increased levels represent higher risk of lipid and glucose disorders (49). In contrast, in the adipose tissue, significant levels of Ces1d translocate onto or get close to the lipid droplets and exert their direct hydrolytic actions to mobilize the droplets. Even though Ces1d also localizes in the ER in the adipose tissue, there is no ApoB expression and hence no VLDL assembly machinery locally in the adipose tissue. To support the notion, the level of VLDL did not change in the Ces1d FKO mice. Therefore, the major function of Ces1d is to immobilize stored lipids in adipose tissue. During diet-induced obesity development, the lipolytic function of Ces1d may release the nutritional burden, which hence improves the whole-body metabolism. In agreement with our observation, recent findings suggest that lipolysis by other lipases triggers a whole-body insulin response that is essential for positive energy expenditure (12, 50).

The mouse Ces1d and its human orthologue CES1 contain functional HVEL or HIEL sequence at their extreme C terminus, respectively. The sequences match the HXEL motif, which is a retrieval sequence necessary and sufficient for ER retention (51). We compared the coding sequences of Ces1d in the liver and adipose tissue and we did not find any difference. Then, what factors drive the translocation of Ces1d onto the cytosolic lipid droplets in the adipose tissue? Glycosylation modifications in Ces1d are correlated with its ER localization (52). However, comparison of the samples side-by-side suggested similar glycosylation levels in both liver and adipose tissue. Interestingly, in our previous report, we found that significantly more Ces1d co-localizes with lipid droplets in adipose tissues in response to cold stimulation, suggesting the β-adrenergic signaling-stimulated cAMP-PKA pathway might be involved in the translocation from the ER to the lipid droplets (12). Nevertheless, the detailed molecular mechanisms governing the translocation of Ces1d warrant further elucidation.

Previously, Ces1d was reported to be a lipase of fat cake extracts in vitro (46). The current study, together with our previous results, further demonstrates its lipase function in vivo (12). Intriguingly, when compared with a classic lipase ATGL, Ces1d produces higher levels of short-chain fatty acids and some long-chain unsaturated fatty acids 18:3n6. Of note, these short-chain fatty acids might

further be proceeded to form long-chain fatty acids by conjugation in vivo (53). Under cold stimulation, the FFAs produced by Ces1d in adipose tissue are used as substrates for accelerated β-oxidation and enhanced thermogenesis (54, 55). Indeed, we observed impaired energy expenditure in the mice when Ces1d function is blocked pharmacologically or genetically (12). Our lipidomic results revealed that deficiency of Ces1d led to not only accumulation of TG, but also decreased levels of PA and other lipids. Further analysis of the lipidomic data allowed us to build the lipid metabolic pathways that may explain the changed lipid profiles, such as increased levels of DG, PG, CL, SM, and PIP as well as decreased levels of PA, Cer, and PI in circulation. These changes reflect the complexity of the functions of Ces1d on lipid metabolism in vivo.

We found the morphology and numbers of the lipid droplets in adipose tissue of FKO mice were aberrantly changed. The effect of loss of Ces1d on lipid droplet dynamics could be in two aspects: first, the deficiency of lipolysis due to lack of Ces1d may induce abnormal accumulation of lipids in the droplets. In that way, the sizes of the lipid droplets are increased; second, the rate of lipid transfer into existing cytosolic lipid droplets is significantly declined in the absence of Ces1d. Therefore, the growth (maturation) of the lipid droplets is delayed (56, 57). As the result, the numbers of the lipid droplets decrease. To support the hypothesis, we found the lipid droplets in the FKO mice were larger with less numbers. Moreover, we showed here that PlIN2 and PlIN3, the markers for nascent lipid droplets (58, 59), were dramatically decreased in the adipose tissue of the FKO mice. Previously, we demonstrated that loss-of-function of Ces1d leads to decreased transcriptional activity of PPARγ (12). This could be the underlined mechanism governing the decreased levels of PlIN2 and PlIN3 given that their expressions are directly regulated by PPARγ (60).

Peripheral tissues gain access to the lipid energy reserves from adipose tissue predominantly via the circulating FFAs produced by lipolysis (61). Mitochondrial FFA oxidation (FAO) plays a fundamental role in burning the FFAs to fulfill the energy supply (42, 62, 63). Previous study has demonstrated that lack of functions of ATGL or HSL disrupted mitochondrial respiration, which was associated with a decreased oxidative phosphorylation (OxPhos), ultimately leading to the dysfunction of mitochondria (64). In line with the previous findings, we found that inhibition of Ces1d with WWL-229, a specific Ces1d inhibitor significantly impaired mitochondrial functions (12). In the current study, by using mRNA-Seq approach, we further revealed that among all the down-regulated genes in the adipose tissue of FKO mice, most are related to mitochondrial

*P < 0.05, n.s, no significance. **(D)** Analysis of luciferase reporter activity for HNF4α in HepG2 cells treated by the serum from WT or FKO mice. RLU, relative light units (n = 6 per group, each point represents a biology replicate, representative of three repeats). Data are represented as mean ± SEM, *t* test, **P < 0.01. **(E)** qPCR analysis for the mRNA levels of *Srebf1*, *Srebf2*, *Mlxipl*, and *Nr1h3* in the liver of WT and FKO mice fed on regular chow or HFD (n = 5 per group, each point represents a biology replicate, representative of three repeats) Data are represented as mean ± SEM, *one-way ANOVA*. **(F)** qPCR analysis for *Creb1* mRNA in the sWAT of WT and FKO mice fed on regular chow or HFD (n = 5 per group, representative of three repeats). Data are represented as mean ± SEM, *one-way ANOVA*. **(G)** WB analysis of pCREB and CREB in the lysates from the sWAT of WT and FKO mice fed on HFD. β-Actin was used as loading control (n = 3 per group, representative of three repeats). **(G, H)** Quantification of the band intensity for pCREB/CREB ratio in (G) (n = 3 per group, each point represents a biology replicate). Data are represented as mean ± SEM, *t* test, *P < 0.05. **(I)** WB analysis of pCREB and CREB in the lysates from the BAT of WT and FKO mice fed on HFD. β-Actin was used as loading control (n = 4 per group, representative of three repeats). **(I, J)** Quantification of the band intensity for pCREB/CREB ratio in (I) (n = 4 per group, each point represents a biology replicate). Data are represented as mean ± SEM, *t* test, *P < 0.05. **(K)** WB analysis of pCREB and CREB in the lysates from the liver of WT and FKO mice fed on HFD. β-Actin was used as loading control (n = 4 per group, representative of three repeats). **(K, L)** Quantification of the band intensity for pCREB/CREB ratio in (K) (n = 4 per group, each point represents a biology replicate). Data are represented as mean ± SEM, *t* test, **P < 0.01.

**A**

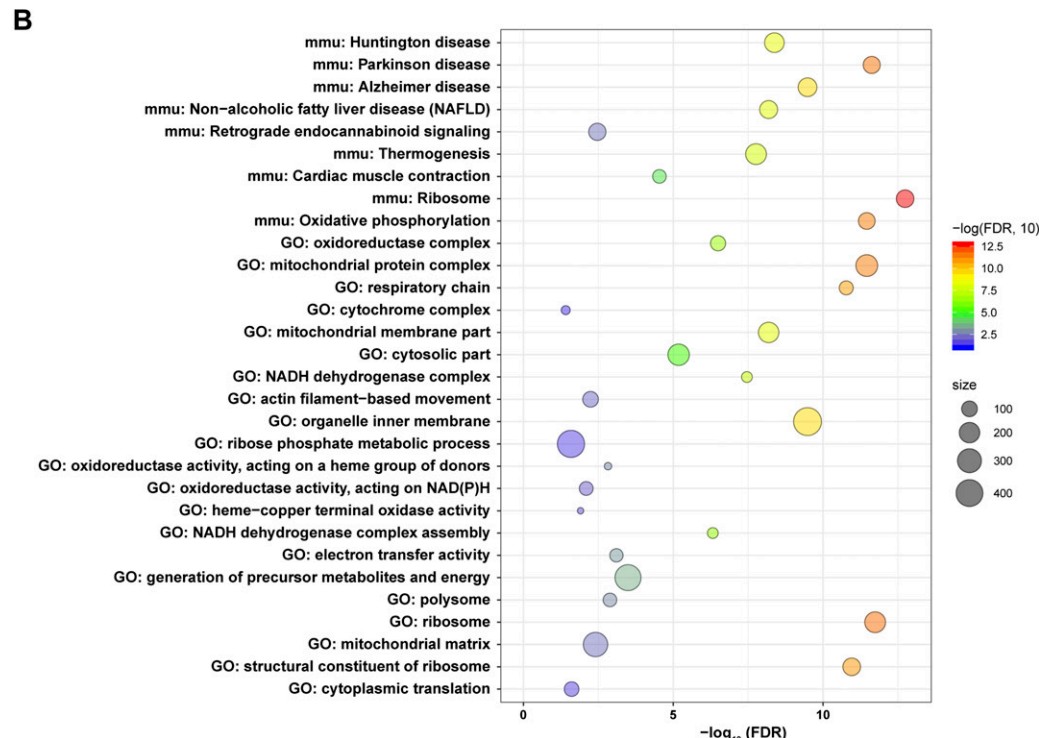

**B**

**C**

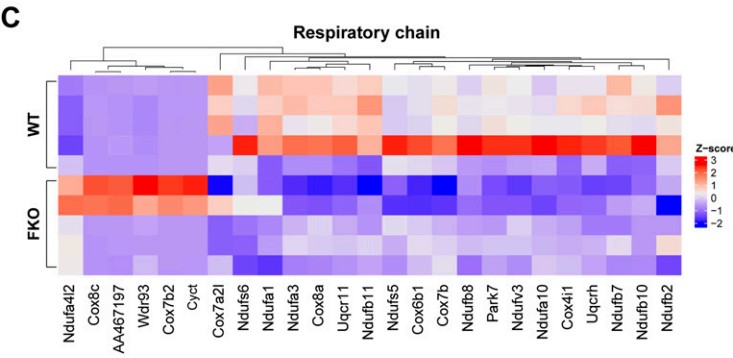

**Figure 10. Deficiency of Ces1d in adipose tissue leads to down-regulation of mitochondrial function related genes.**
**(A)** Volcano plot showing differential expressed genes from RNA-seq data of white adipose tissue from WT versus FKO mice fed on HFD. The red dots denote the significantly differentially expressed genes (DEGs) with absolute log2 (fold change) greater than 1.2 and $-\log_{10}$ false discovery rate (FDR) great than 0.70 (n = 5 per group). **(B)** Bubble chart of the Gene Ontology and KEGG pathways enriched in the down-regulated DEGs (n = 559 genes). Bubble color reflects enrichment strength ($-\log_{10}$ FDR). Bubble size reflects the number of genes in each term. FDR: false discovery rate. **(C)** Heat map showing expression pattern of the respiratory chain related genes in the WAT from WT and FKO mice. Red and blue colors are proportional the high and low expression, respectively.

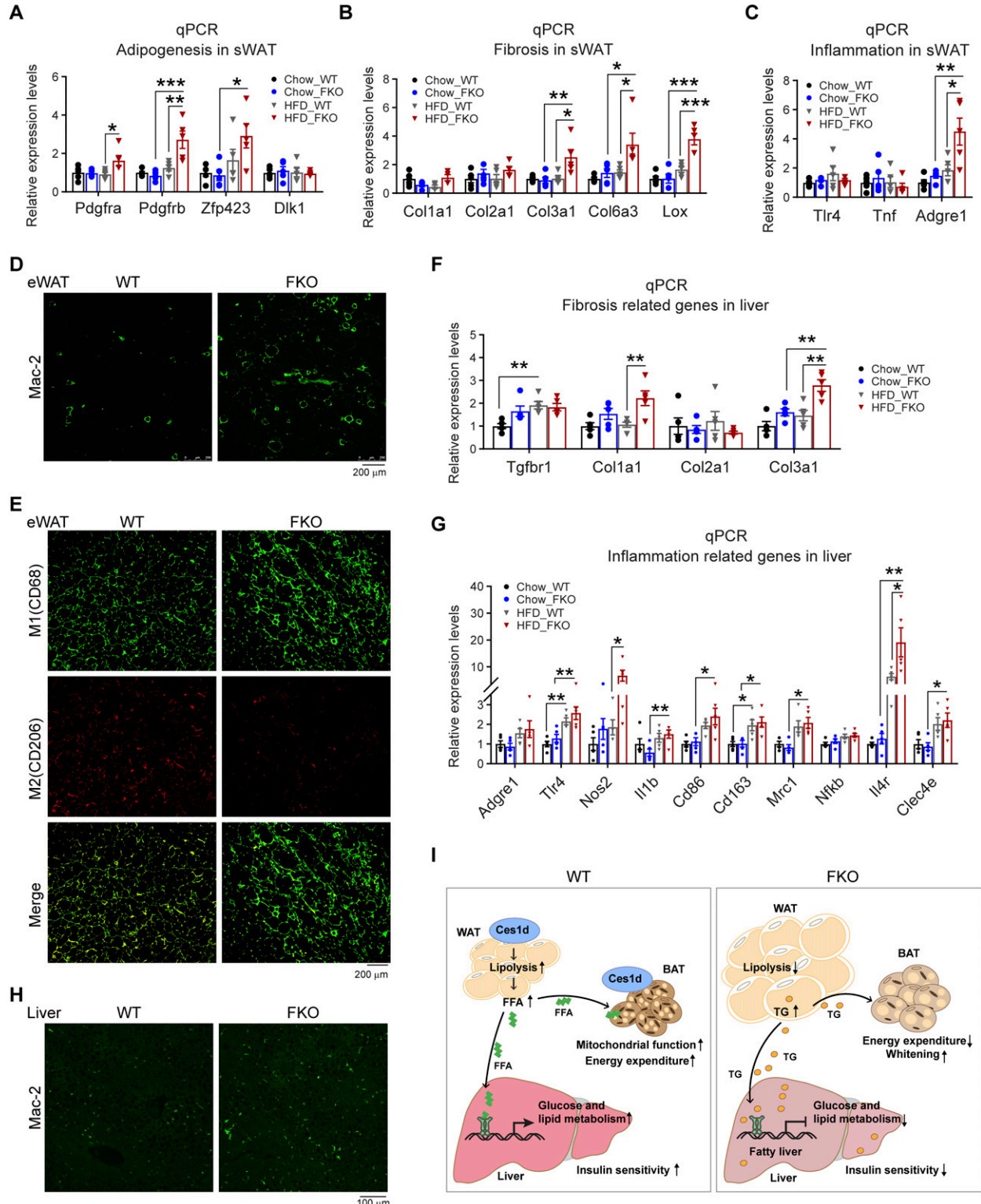

**Figure 11.  Deficiency of Ces1d in adipose tissue causes local unhealthy microenvironment in adipose tissue and the liver, ultimately leading to systemic insulin resistance.**
**(A)** qPCR analysis for the mRNA levels of adipogenesis-related genes including *Pdgfa*, *Pdgfb*, *Zfp243*, and *Dlk1* in the sWAT of WT and FKO mice fed on regular chow or HFD (n = 5 in each group, each point represents a biology replicate, representative of three repeats). Data are represented as mean ± SEM, *one-way ANOVA*, *P < 0.05, **P < 0.01, ***P < 0.001. **(B)** qPCR analysis for the mRNA levels of fibrosis related genes including *Col1a1*, *Col2a1*, *Clo3a1*, *Clo6a3*, and *Lox* in the sWAT from WT and FKO mice fed on regular chow or HFD (n = 5 in each group, each point represents a biology replicate, representative of three repeats). Data are represented as mean ± SEM, *one-way ANOVA*,

functions, such as the factors involved in OxPhos, mitochondrial respiratory chain, and mitochondrial protein complexes. Our studies thus support the concept that proper flux of FFAs are key to maintain the mitochondrial functions.

In addition to provide substrates for β-oxidation to provide energy and for lipid synthesis, the FFAs produced by lipolysis in adipose tissue have been demonstrated to serve as "third messenger" to activate lipid-sensing nuclear receptors (34, 37, 65, 66, 67). For example, ATGL-mediated production of FFAs may activate HNF4α which further up-regulate energy expenditure–related genes in the liver (37, 39). Here, we found that Ces1d hydrolyses the lipids to produce C18 linoleic acids which may bind and hence activate HNF4α. Deficiency of HNF4α leads to lipid metabolic disorders and severe liver steatosis (68, 69, 70). Here, we demonstrated that in the liver of the FKO mice, even though HNF4α levels did not alter, its downstream genes, such as Apoc3, Pek1, and Cyp7a1 were significantly down-regulated. We further confirmed the down-regulations of the target genes are directly via HNF4α suppression in HepG2 cells. Indeed, treatment of the serum from the KFO mice not only led to dramatic down-regulations of HNF4α target genes, but also significant suppression of the luciferase reporter activity specifically driven by HNF4α when compared with the serum from the WT mice. The results support a concept that the adipose tissue–liver crosstalk may be tightly controlled by the adipose tissue–derived FFAs (71, 72, 73, 74). Eventually, the upshot of liver fat accumulation, alongside the impaired enzymatic pathways, eventually leads to fatty liver.

Intriguingly, we observed suppressions of several other transcriptional factors involved in the lipid and glucose metabolism in the FKO mice. These changes reflect a propounding regulation of Ces1d at multiple levels. Among them, the most dramatic change is the post-translational modification of the cAMP-responsive transcription factor CREB in the FKO mice. CREB is an essential regulator for gluconeogenesis, lipolysis, and fatty acid oxidation and its phosphorylation by PKA enhances its transcriptional activities on the related genes (75, 76, 77). In the present study, we revealed that even though its total levels did not change, its phosphorylation levels by PKA were significantly reduced in the adipose tissue and the liver of the FKO mice. The decreased phosphorylation of CREB at least partially explains the down-regulations of Pgc1, Pck1, and G6pc and the consequential glucose and lipid dysregulations in the FKO mice. How the phosphorylation of CREB is attenuated in the

FKO mice needs to be further investigated. Interestingly, phosphorylation of ACC was significantly increased in the liver and adipose tissue of the FKO mice. Given that phosphorylation of ACC by AMPK inhibits its catalytic function on de novo lipid synthesis (78, 79), this enhanced phosphorylation effect might be considered as a compensatory reaction in response to the increased lipid flux to the organs/tissues in the Ces1d FKO mice.

The excessive lipid accumulation caused by lipolytic deficiency leads to hypotrophy and hyperplasia in the adipose tissue (2, 74, 80). Indeed, we found that in addition to larger lipid droplets, adipogenic genes were induced in the FKO mice upon HFD feeding. Moreover, the rapidly expanded adipose masses induces local hypoxia conditions, which further trigger pro-fibrotic and pro-inflammatory reactions in the adipose tissue (2, 81, 82, 83, 84). In agreement with this concept, we found the pro-fibrotic and pro-inflammatory genes were significantly up-regulated in the adipose tissue of the FKO mice upon HFD challenge. As a result, the pro-inflammatory M1-macrophages accumulated in the eWAT of the FKO mice. The upshot of hepatic lipid accumulation, along with other factors abnormally circulated into the liver from the unhealthy adipose tissue, might further result in pro-inflammatory and pro-fibrotic shift in the liver (80, 85). Indeed, we found up-regulations of the pro-inflammatory and pro-fibrotic genes as well as macrophage accumulation in the liver of the FKO mice.

In conclusion, in the present study, we identified Ces1d as a major lipid hydrolase in adipose tissue. Ces1d exerts a distinct function on metabolic regulations in adipose tissue. Deficiency of Ces1d causes both glucose and lipid metabolic disorders in adipose tissue and the liver. Consequently, the FKO mice develop lipotoxicity, inflammation and insulin resistance. Our findings unveil a previously underappreciated aspect of lipolytic signaling mediated by Ces1d, thus highlighting its therapeutic potential to treat obesity and related metabolic disorders.

# Materials and Methods

### Animals and HFD studies

All of the animal studies were reviewed and approved by the Animal Welfare Committee (AWC) of University of Texas Health Science Center at Houston (Animal protocol number: AWC-21-0019). Wild-

---

*P < 0.05, **P < 0.01, ***P < 0.001. **(C)** qPCR analysis for the mRNA levels of inflammation related genes including *Tlr4*, *Tnf*, and *Adgre1* in the sWAT of WT and FKO mice fed on regular chow or HFD (n = 5 in each group, each point represents a biology replicate, representative of three repeats). Data are represented as mean ± SEM, *one-way ANOVA*, *P < 0.05, **P < 0.01. **(D)** IF staining with α-Mac-2 antibody on the eWAT of WT and FKO mice fed on HFD (representative of six fields, experiments were repeated for three times; scale bars: 200 μm). **(E)** Co-IF staining with α-CD68 (green) and α-CD206 (red) antibodies on the eWAT of WT and FKO mice fed on HFD (representative of six fields, experiments were repeated for three times; scale bars: 200 μm). **(F)** qPCR analysis for the mRNA levels of fibrosis related genes including *Tgfbr1*, *Col1a1*, *Col2a1*, and *Col3a1* in the liver from WT and FKO mice fed on regular chow or HFD (n = 5 in each group, each point represents a biology replicate, representative of three repeats). Data are represented as mean ± SEM, *one-way ANOVA*, **P < 0.01. **(G)** qPCR analysis for the mRNA levels of inflammation related genes including *Adgre1*, *Tlr4*, *Nos2*, *Il1b*, *Cd86*, *Cd163*, *Cd206*, *Nfkb*, *Il-4r*, and *Clec4e* in the liver of WT and FKO mice fed on regular chow or HFD (n = 5 in each group, each point represents a biology replicate, representative of three repeats). Data are represented as mean ± SEM, *one-way ANOVA*, *P < 0.05, **P < 0.01. **(H)** IF staining with α-Mac-2 antibody on the liver of WT and FKO mice fed on HFD (representative of six fields, experiments were repeated for three times, scale bars: 100 μm). **(I)** Left: In the normal adipose tissue, Ces1d hydrolyzes the lipid droplet TG. The produced FFAs circulate into BAT and the liver where they serve as the substrates for energy generation. Moreover, the FFAs also function as signaling molecules to activate the lipid-sensing transcriptional factors to regulate the glucose and lipid metabolic homeostasis. Right: In the Ces1d-deficient adipose tissue, insufficiency of lipolysis leads to the formation of larger lipid droplets in adipocytes. The excessive nutritional stress, especially under HFD challenging condition may cause increased ectopic deposition of TG in peripheral tissues. This leads to lipotoxicity, which further induces fatty liver and whitening of the BAT. Concomitantly, absence of the key signaling FFAs might blunt the proper regulations of key metabolic genes, which further worsens the metabolic adverse effects. Ultimately, the mice develop whole-body insulin resistance.

type (WT) C57BL/6J (Stock 000664) and Adiponectin-Cre (Stock 010803) mice were purchased from Jackson Laboratories. The Ces1d-floxed ($Ces1d^{flx/flx}$) mouse was generated by Dr. Grant A Mitchell from the Ste-Justine Hospital in Montreal, Canada, and crossed onto C57BL/6J background (18). 8-wk-old male FKO and their littermate floxed control (from here on referred to as WT) mice were used for all the experiments. Mice were housed in the animal facility (22°C ± 1°C) on a 12-h light/dark cycle with ad libitum access to water and regular chow diet, unless indicated otherwise. For the HFD challenge, the age matched male WT and FKO mice were fed with the HFD (containing 60% calories from fat; Bio-Serv) for 14 or 25 wk. The total body mass, fat mass and lean mass were measured by EchoMRI-100T (Echo Medical Systems).

### Primary mouse hepatocyte isolation and culture

Primary hepatocytes were isolated from the fresh livers of the WT or liver-specific $Hnf4\alpha$ knockout mice following the standard two-step collagenase perfusion technique (68, 86). For HNF4 transcriptional activity assay, the hepatocytes were plated in in 12-well plates. Cells were harvested for qPCR after treated for medium with 10% WT or FKO mice serum for 24 h.

### Cell culture and treatment

HepG2 cells were obtained from American Tissue Culture Collection (ATCC) and were cultured in growth medium EMEM (ATCC, Cat. no. 30-2003) supplemented with 10% fetal bovine serum, 1× antibiotic-antimycotic and 10 mM Hepes in a humidified atmosphere of 5% of $CO_2$ at 37°C. For serum treatment assay, the cells were seeded in the six-well plates. 18 h later when the confluence reached to 75%, the cells were treated with EMEM containing 10% of sera collected from the WT or FKO mice. Upon the serum treatment for 18 h, the cells were harvested for measurements of gene expression levels.

### Luciferase assay

HepG2 cells were co-transfected with the plasmids of HNF4$\alpha$-specific luciferase reporter ApoB-luc and $\beta$-Galactosidase. 24 h upon transfection, the cells were treated with EMEM containing 10% of sera collected from the tail veins of the WT or FKO mice. Upon the serum treatment for 18 h, the cells were harvested for the luciferase assay. Briefly, the luciferase activities in the lysed cells were determined by Dual-Light Luciferase & $\beta$-Galactosidase Reporter Gene Assay System (Thermo Fisher Scientific), and normalized to $\beta$-galactosidase activity.

### Measurement of insulin signaling in vivo

The HFD-fed FKO and their littermate WT control mice were fasted for 6 h. Then, they were i.p. injected with insulin (0.5 U/kg; Sigma-Aldrich). Upon insulin injection for 15 min, the mice were euthanized and the livers, the muscles, BAT, and WAT were immediately collected and frozen for further AKT phosphorylation assay by Western blotting (refer to below for details).

### Western blotting

For Western blot analysis, tissues were lysed in NENT buffer (100 mM NaCl, 20 mM Tris-Cl [pH 8.0], 0.5 mM EDTA, and 0.5% [vol/vol] Nonidet P-40 [NP-40]) with freshly added protease inhibitors. Tissue samples were spun at 12,000$g$ for 10 min at 4°C and the supernatant was extracted. Protein samples were separated by SDS–PAGE and transferred onto polyvinylidene difluoride (PVDF) membrane followed by blocked in 5% fat free milk. Afterward, the PVDF membranes were incubated with primary antibodies at 4°C overnight or RT for 3 h followed by washed with 1 × PBST (0.1% Tween 20 in 1 × PBS, pH = 7.4) for three times (5 min/each wash). Then the membranes were incubated with IRDye 800 CW or 680 RD secondary antibodies (LI-COR) at RT for 1 h. After being washed with 1 × PBST for three times (5 min/each wash), the blots were imaged by Odyssey software (LI-COR Biosciences) and the band densities were analyzed by ImageJ software (NIH). The primary antibodies were: Ces1d (goat $\alpha$-Ces1d, PA5-47802; Invitrogen; mouse $\alpha$-Ces1d antibody, MA5-24244; Invitrogen), PDI (MA3–019; Invitrogen), ERK1/2 (9102; Cell Signaling Technology), PLIN1 (ab61682; Abcam), PLIN2 (ab52356; Abcam), PLIN3 (ab47639; Abcam), PLIN5 (ab228111; Abcam), BiP (PA1–014A; Thermo Fisher Scientific), Nogo (sc-271878; Santa Cruz Biotechnology), HSL (4107; Cell Signaling Technology), pHSL (Ser660) (4126; Cell Signaling Technology), ATGL (sc-36527; Santa Cruz Biotechnology), CGI58 (sc-365278; Santa Cruz Biotechnology), SCD1 (2438; Cell Signaling Technology), PPAR$\gamma$ (sc-72737; Santa Cruz), ACC1 (4190; Cell Signaling Technology), pACC1 (3661; Cell Signaling Technology), CREB (9197; Cell Signaling Technology), pCREB (9198; Cell Signaling Technology), AKT (4691; Cell Signaling Technology), pAKT (4060; Cell Signaling Technology), HNF4a (PP-K9218-00; R&D System), $\beta$-Actin (61256; BD Biosciences), $\alpha$-tubulin (2144; Cell Signaling Technology), and GAPDH (2118; Cell Signaling Technology).

### Removal of glycosylation modification in Ces1d

For de-glycosylation assay, the lysates from the liver, sWAT, and BAT were pretreated with protein de-glycosylation mix II (P6044S; New England BioLabs) or reaction buffer only according to the manufacturer's instructions. Then the samples were boiled for 5 min together with the protein loading buffer and loaded on SDS gels for Western Blotting on Ces1d.

### RNA preparation and quantitative PCR (qPCR)

Total RNA from the tissues or HepG2 cells were isolated by using Trizol reagent (15596018; Thermo Fisher Scientific) following the manufacturer's instructions. For qPCR, cDNAs were obtained by reverse-transcribing 1 $\mu$g of total RNAs with RevertAid Reverse Transcription Kit (K1691; Thermo Fisher Scientific). qPCR reactions were carried out on Bio-Rad CFX96 system (Bio-Rad Laboratories). Results were normalized by $\beta$-Actin and calculated using the $2^{-\Delta\Delta Ct}$ method (12). All of the primer sequences for the mouse study were described in our previous publication (12). All of the primer sequences for the human HNF4$\alpha$ targeting genes were described previously (87, 88, 89, 90, 91). The primers were listed in the Table S3 (12, 92).

## RNA-seq and data analysis

Total mRNA samples were isolated from the eWAT using Trizol reagent as described above, and then they were performed RNA-seq assay in Cancer Genomics Center at The University of Texas Health Science Center at Houston. The quality of the total RNAs was checked using Agilent RNA 6000 Pico kit (#5067-1513) by Agilent Bioanalyzer 2100 (Agilent Technologies). RNA with an integrity number of greater than 7 was used for library preparation. Libraries were prepared with KAPA mRNA HyperPrep (KK8581; Roche) and KAPA UDI Adapter Kit 15 $\mu$M (KK8727; Roche) following the manufacturer's instructions. The quality of the final libraries was examined using Agilent High Sensitive DNA Kit (#5067-4626) by Agilent Bioanalyzer 2100, and the library concentrations were determined by qPCR using Collibri Library Quantification kit (#A38524500; Thermo Fisher Scientific). The libraries were pooled evenly and applied for the paired-end 75-cycle sequencing on an Illumina NextSeq 550 System (Illumina, Inc.) using High Output Kit v2.5 (#20024907; Illumina, Inc.). We used ultrafast universal RNA-seq aligner STAR (v2.5.3a) to conduct the read alignment to mouse reference genome GRCm38. Gene read counts were calculated through setting the argument–quantMode to "GeneCounts" to let STAR count number of uniquely mapped reads per gene from the GencodeM15 reference. We filtered out those genes with <5 reads in all samples and conducted the differential expression analysis by DESeq2 (93). The $P$-values of genes were adjusted using the Benjamin and Hochberg's procedure to control the false discovery rate. The differentially expressed genes were defined as the genes with fold change >1.2 and false discovery rate <0.2. Overrepresented analysis for non-redundant Gene Ontology, KEGG pathway, and Gene Set Enrichment Analysis (94) for the WikiPathways/Gene Ontology and KEGG pathways was performed using WebGestalt (v0.4.3) package in R (95). All raw data and processed read count have been submitted to GEO (GSE173893).

## Tissue histology

Tissues were collected immediately after the mice were euthanized. They were fixed in 10% PBS-buffered formalin (pH = 7.4; Thermo Fisher Scientific) at RT for 48 h. The tissues were then paraffin-embedded and sectioned at 5-$\mu$m size. After being deparaffinized, the sections were stained with hematoxylin and eosin (H & E; Sigma-Aldrich) using a standard protocol as described previously (12). For immunofluorescence (IF) staining, the deparaffinized sections were permeabilized with 1× PBS containing 0.2% Triton X-100 for 10 min followed by incubated with sodium citrate buffer at 95°C for 30 min for antigen retrieval. After being blocked in 5% bovine serum albumin, the sections were stained with primary antibodies at 4°C overnight followed by washed with 1 × PBST for three times (5 min/wash). The sections were then incubated with secondary antibodies at RT for 1 h. After washing with 1 × PBST for three times (5 min/wash), the sections were mounted for imaging under the Leica TCS SP5 Confocal Laser Scanning Microscope. The primary antibodies used were those described above and CD206 (AF2534; R&D Systems) and CD68 (MA5-13324; Invitrogen). The secondary antibodies used are Alexa Fluor 488–conjugated donkey anti-rabbit IgG and Alexa Fluor 647–conjugated donkey anti-goat IgG (Jackson ImmunoResearch Laboratories).

## Indirect calorimetric measurements

The metabolic cage studies were performed by TSE cage chamber (TSE Systems). Briefly, the FKO and their littermate control mice were housed individually in the cage chambers under a 12-h light/dark cycle (light on between seven a.m and seven p.m). The mice are free to access water and the HFD during the assays. The profile for energy expenditure, including $O_2$ consumption, $CO_2$ production, and heat generation, were documented continuously. The data were calculated by normalizing to their body weights (96).

## IPGTTs and ITTs

An i.p. glucose tolerance test (IPGTT) and an ITT were performed as previously described (83). In brief, for the IPGTT, the mice were fasted for 5 h before being injected with glucose (2.5 g/kg body weight) via i.p. At the indicated time points, tail blood samples were collected with heparin-coated capillary tubes and centrifuged at 4,427$g$ for 6 min for serum purification. The glucose levels in the serum were measured by the glucose oxidase/peroxidase method (97). For the ITT, the mice were fasted for 4 h before being injected with insulin (0.5 U/kg body weight; Sigma-Aldrich, I9278) via i.p. At the indicated time points, tail blood samples were collected with heparin-coated capillary tubes and centrifuged at 4,427$g$ for 6 min for the serum purification. The glucose levels in serum were measured as previously described (97).

## Plasma and liver biochemistry

Enzymatic assays kits were used to perform the plasma and hepatic chemistries: insulin (Cat. no. 90080; Crystal Chem), triglyceride (TAG) (Cat. no. ab65336; Abcam), free fatty acid (FFA) (Cat. no. EFFA-100; Bioassay System), glycerol (Cat. no. EGLY-200; Bioassay System), cholesterol (Cat. no. 10007640; Cayman Chemical), high-density lipoprotein (HDL) cholesterol, very-low-density lipoprotein (VLDL) cholesterol (Cat. no. ab65390; Abcam), FGF21 (Cat. no. ab212160; Abcam), leptin (Cat. no. ab100718; Abcam), and $\beta$-Hydroxybutyrate (Cat. no. MAK041; Sigma-Aldrich). The measurements were performed according to the manufacturer's instructions.

## In vitro hydrolysis assay

Total lipids were extracted from the sWAT of 6-wk-old male wild-type C57BL/6J mice using lipid extraction buffer chloroform/methanol/water (2/1/0.6, vol/vol/v) as previously described (98). Briefly, to prepare the total lipids, the sWAT was homogenized in the lipid extraction buffer and kept shaking for 1 h at RT. After centrifuged at 12,000$g$ for 15 min, the clear organic phase was collected and dried under nitrogen in a flame hood. Then the lipids were emulsified in 100 $\mu$l of 100 mM potassium phosphate buffer (pH 7.0) on ice by sonication. His-tagged ATGL and Ces1d proteins were expressed in HEK293 cells and purified with TALON Metal Affinity Resin (Clontech Laboratories, Inc) revised from the

guidance by the manufacturer. To avoid the purified His-tagged proteins form complexes with other enzymes in the cell lysates, the denaturing approach was applied. Briefly, the cell lysates in the denaturing buffer (pH 8.0) containing 6 M guanidine-HCL were incubated with TALON metal beads at 4°C for 4 h. Then the resin beads were spin down and washed three times with the washing buffer (pH 7.0) containing 6 M guanidine-HCL and 25 mM imidazole. Upon the last wash, the beads were maintained in the equilibration buffer (pH 8.0, 1:1 ratio). 25 $\mu$l of the beads were removed to check the purity of the proteins on SDS gel. For the in vitro hydrolysis reaction, 50 $\mu$g of the proteins (binding with resin beads) were mixed with 200 $\mu$l of the sonicated lipids in the Eppendorf tubes and the tubes were kept shaking at 37°C for 1 h. Then the reaction products were extracted with chloroform/methanol (2:1, vol/vol) mixture and the organic phase was separated by TLC (98). To determine the fatty acid composition, the gel slices of the TCL plate where the fatty acid fraction located were purified and sent for LC-MS/MS analysis.

### Profiling of fatty acids by LC-MS/MS

FFA profiling was performed at the Metabolomics Core at MD Anderson Cancer Center, using a chemical derivatization approach. An internal standard mixture consisted of 12.5 $\mu$g/ml of (1, 2, 3, 4, 5, and 6-$^{13}C_6$) 22:0 and 25 $\mu$g/ml $^{13}$C-labeled 14:0, 16:1n7c, 16:0, 17:0, 18:2n6, 18:1n9c, 18:1n9t, and 18:0 in ethanol (Cambridge Isotope Laboratories). Fatty acids were extracted from TLC gel slice samples by combining samples with 32 $\mu$l internal standard mixture and 1 ml extraction solvent (methanol) and vortexing for 5 min. After centrifugation at 4,122$g$ at 4°C for 10 min, the supernatants were transferred to 2-ml vials with Teflon caps and dried using a centrifugal vacuum concentrator. Extracted FFAs were converted to acyl chloride intermediates by treatment with 200 $\mu$l 2 M oxalyl chloride in dichloromethane at 65°C for 5 min. The solutions were then dried and samples were derivatized by adding 150 $\mu$l of 1% (vol/vol) 3-picolylamine in acetonitrile. Finally, the solutions were dried and stored at −80°C. Derivatization products were reconstituted in 100 $\mu$l ethanol, transferred to autosampler vials, dried, and then reconstituted in 15 $\mu$l ethanol. Injection volume was 5 $\mu$l. Mobile phase A (MPA) was 0.1% formic acid in water and mobile phase B (MPB) was 0.1% formic acid in acetonitrile. The chromatographic method included a Thermo Fisher Scientific Accucore C30 column (2.6 $\mu$m, 150 × 2.1 mm) and the following gradient elution: 0–5 min, 65% MPB; 5–5.1 min, 65–90% MPB; 5.1–55 min, 90% MPB; 55–55.1 min, 90–65% MPB; 55.1–60 min, and 65% MPB. A Thermo Fisher Scientific Orbitrap Fusion Tribrid mass spectrometer with heated electrospray ionization source was operated in data-dependent acquisition mode with a scan range of 150–550 m/z.

### Lipidomics

For the lipidomic assay, tail blood samples were collected from 5-h fasting mice with heparin-coated capillary tubes and centrifuged at 6,000 rpm (F45-21-11 rotor, Eppendorf) for 6 min for serum purification. All the mice were fed with HFD for 14 wk. 120 $\mu$l of the sera were sent for assay. To each 100 $\mu$l sample, 5 $\mu$l of Avanti SPLASH LIPIDOMIX Mass Spec Standard (330707) in methanol was added

followed by 3 $\mu$l of 10 mM butylated hydroxytoluene in methanol and 242 $\mu$l of −80°C ethanol. The tubes were vortexed 5 min, then centrifuged at 4°C for 10 min at 17,000$g$. The supernatants were then transferred to a Phenomenex Impact Protein Precipitation Plate (CE0-7565) and filtered through using a vacuum manifold. The pellets resulting from centrifugation were re-extracted with 300 $\mu$l of ethanol and the supernatants were once again passed through the protein precipitation plate. The plate wells were then rinsed with 200 $\mu$l ethanol to elute residual lipids. The lipid extracts were transferred to new Simport tubes, prewashed with methanol, and 200 $\mu$l ethanol was used to wash the collection wells of the plate. The wash solutions were combined with the extracts and a centrifugal vacuum concentrator was used to dry the samples.

Dried samples were reconstituted in 100 $\mu$l ethanol. The injection volume was 15 $\mu$l. Mobile phase A (MPA) was 40:60 acetonitrile: 0.1% formic acid in 10 mM ammonium acetate. Mobile phase B (MPB) was 90:8:2 isopropanol: acetonitrile: 0.1% formic acid in 10 mM ammonium acetate. The chromatographic method included a Thermo Fisher Scientific Accucore C30 column (2.6 $\mu$m, 150 × 2.1 mm) maintained at 40°C, autosampler tray chilling at 8°C, a mobile phase flow rate of 0.200 ml/min, and a gradient elution program as follows: 0–7 min, 20–55% MPB; 7–8 min, 55–65% MPB; 8–12 min, 65% MPB; 12–30 min, 65–70% MPB; 30–31 min, 70–88% MPB; 31–51 min, 88–95% MPB; 51–53 min, 95–100% MPB; 53–60 min, 100% MPB; 60–60.1 min 100–20% MPB; and 60.1–70 min, 20% MPB.

A Thermo Fisher Scientific Orbitrap Fusion Lumos Tribrid mass spectrometer with heated electrospray ionization source was operated in data dependent acquisition mode, in both positive and negative ionization modes, with scan ranges of 150–700 and 675–1,500 m/z. An Orbitrap resolution of 120,000 (FWHM) was used for MS1 acquisition and a spray voltage of 3,600 and −2,900 V were used for positive and negative ionization modes, respectively. Vaporizer and ion transfer tube temperatures were set at 275 and 300°C, respectively. The sheath, auxiliary, and sweep gas pressures were 35, 7, and 0 (arbitrary units), respectively. For MS2 and MS3 fragmentation, a hybridized HCD/CID approach was used. Each sample was analyzed using 4 × 10 $\mu$l injections making use of the two aforementioned scan ranges, in both ionization modes. Data were analyzed using Thermo Fisher Scientific LipidSearch software (version 4.2.23) and R scripts written in house.

### Isolation of proteins from cytoplasm, ER, and lipid droplets

Isolation of ER and lipid droplets was carried out following a previous publication (12). Briefly, the liver and adipose tissues were homogenized in 2 ml of buffer A (20 mM tricine and 250 mM sucrose, pH 7.8) with 0.2 mM PMSF and sequentially centrifuged at a speed of 1,000$g$ to remove cell debris as well as nuclear, and then, centrifuged at 12,000$g$ to remove mitochondria. The clear supernatant together with the top lipid layer was carefully collected and subjected to ultra-centrifugation at speed of 1,000,00$g$ for 1 h to isolate ER at the bottom fraction. The middle supernatant fraction was collected as cytosol part. The top lipid layer was collected as the lipid droplets part. To further remove possible contaminations on the lipid droplet surface, the lipid layer was carefully collected and washed with 2 ml buffer B (20 mM Hepes, 100 mM KCl, and 2 mM MgCl$_2$, pH 7.4), followed with ultra-centrifuging at speed of 1,000,00$g$

for 1 h twice. The lipid droplet surface proteins were extracted by chloroform-acetate method as previously described (12).

## Statistical analysis

All data were represented as mean ± SEM. Statistical analyses were performed using GraphPad Prism 8 software (Graph Pad Software Inc). An unpaired *t* test was used for statistical significance analysis when two groups were compared. One-way ANOVA was used for comparison among multiple groups. A *P*-value less than 0.05 was considered statistically significant.

# Data Availability

The RNA sequencing data have been deposited to GEO under the accession number: GSE173893 (https://www.ncbi.nlm.nih.gov/geo/query/acc.cgi?acc=GSE173893).

# Supplementary Information

# Acknowledgements

The authors would like to thank their colleagues in the Center of Metabolism and Degenerative Diseases for technical support and critical discussions. We thank Dr. Frances M Sladek at the University of California, Riverside, for the ApoB-Luc reporter construct. We also thank Dr. Zhengmei Mao in the microscopy core of the Institute of Molecular Medicine for assistance on imaging and tissue processing. The *Ces1d* floxed mice were made by Dr. Grant A Mitchell from the Division of Medical Genetics, Ste-Justine Hospital in Montreal, Canada. This study was supported by the National Institute of Health (NIH) grants R01DK109001, R56DK124419 (to K Sun), R01DK092590 (to R Berdeaux), R01DK 125922 (to K Eckel-Mahan); Canadian Institutes of Health Research grant PS 156314 (to R Lehner) and grants S10OD012304-01, P30CA016672 (to PL Lorenzi). This work is supported in part by the Clinical and Translational Proteomics Service Center at the University of Texas Health Science Center. The RNA sequencing and analyses were performed by Cancer Genomics Core supported by the UTHealth Cancer Prevention and Research Institute of Texas (CPRIT RP180734).

## Author Contributions

G Li: data curation, formal analysis, validation, investigation, and methodology.
X Li: conceptualization, data curation, formal analysis, validation, investigation, methodology, and writing—review and editing.
L Yang: data curation, validation, investigation, and methodology.
S Wang: data curation, formal analysis, validation, and investigation.
Y Dai: resources, software, validation, and methodology.
B Fekry: data curation, investigation, and methodology.
L Veillon: resources, methodology, and writing—review and editing.
L Tan: resources, data curation, software, investigation, and methodology.
R Berdeaux: resources, formal analysis, and writing—review and editing.
K Eckel-Mahan: resources, formal analysis, methodology, and writing—review and editing.
PL Lorenzi: resources, software, methodology, and writing—review and editing.
Z Zhao: software, formal analysis, and writing—review and editing.
R Lehner: conceptualization, resources, supervision, and writing—review and editing.
K Sun: conceptualization, data curation, formal analysis, supervision, funding acquisition, validation, investigation, visualization, methodology, project administration, and writing—original draft, review, and editing.

## Conflict of Interest Statement

The authors declare that they have no conflict of interest.

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
