## [Reviewer comments · Life Science Alliance]

Adipose tissue-specific ablation of *Ces1d* causes metabolic dysregulation in mice

Gang Li, Xin Li, Li Yang, Shuyue Wang, Yulin Dai, Baharan Fekry, Lucas Veillon, Lin Tan, Rebecca Berdeaux, Kristin Eckel-Mahan, Philip L. Lorenzi, Zhongming Zhao, Richard Lehner, and Kai Sun

DOI: 10.26508/lsa.202101209

Corresponding author(s): Kai Sun (The University of Texas Health Science Center at Houston)

Review timeline:

Submission Date:	2021-08-23
Editorial Decision:	2021-09-22
Revision Received:	2022-03-10
Editorial Decision:	2022-03-29
Revision Received:	2022-04-08
Accepted:	2022-04-11

Scientific Editor: Eric Sawey

Transaction Report:

No Peer Review Process File is available with this article, as the authors have chosen not to make the review process public in this case.

Re: Life Science Alliance manuscript #LSA-2021-01209-T

Dr. Kai Sun
University of Texas Health Science Center at Houston

Dear Dr. Sun,

Thank you for submitting your manuscript entitled "Adipose Tissue Specific Carboxylesterase 1d (Ces1d) Mediates Whole-Body Metabolic Homeostasis" to Life Science Alliance. The manuscript was assessed by expert reviewers, whose comments are appended to this letter. We invite you to submit a revised manuscript addressing the Reviewer comments.

Thank you for this interesting contribution to Life Science Alliance. We are looking forward to receiving your revised manuscript.

Sincerely,

-- High-resolution figure, supplementary figure and video files uploaded as individual files: See our detailed guidelines for preparing your production-ready images,

<https://www.life-science-alliance.org/authors>

B. MANUSCRIPT ORGANIZATION AND FORMATTING:

RE: Life Science Alliance Manuscript #LSA-2021-01209-TR

Dr. Kai Sun
The University of Texas Health Science Center at Houston
1825 Pressler ST
Houston 77030

Dear Dr. Sun,

Thank you for submitting your revised manuscript entitled "Adipose tissue-specific ablation of *Ces1d* causes metabolic dysregulation in mice". We would be happy to publish your paper in Life Science Alliance pending final revisions necessary to meet our formatting guidelines.

- please upload your main manuscript text as an editable doc file
- please upload your main and supplementary figures as single files
- please note that figures should be provided as one figure per file
- please upload your Tables in editable .doc or excel format
- please add the Twitter handle of your host institute/organization as well as your own or/and one of the authors in our system
- please add a Category for your manuscript in our system
- please use Capital letters when introducing panels in Figures, their legends, and callouts in the manuscript text
- there are two panels labelled "D" in Figure 1, please correct
- please add your main, supplementary figure, and table legends to the main manuscript text after the references section
- please consult our manuscript preparation guidelines <https://www.life-science-alliance.org/manuscript-prep> and make sure your manuscript sections are in the correct order
- please add an Author Contributions section to your main manuscript text
- please add a conflict of interest statement to your main manuscript text
- please add a Data Availability section to mention the RNA-seq GEO accession info again
- the References found in Table S1 should be included in the main References section, not separately underneath the Table

FIGURE CHECKS:

- missing scale bars for figure 1J
- the beta-actin blot in figure 1, panel O is hard to read

You will be guided to complete the submission of your revised manuscript and to fill in

all necessary information. Please get in touch in case you do not know or remember your login name.

A. FINAL FILES:

B. MANUSCRIPT ORGANIZATION AND FORMATTING:

Thank you for this interesting contribution, we look forward to publishing your paper in

Life Science Alliance.

Sincerely,

RE: Life Science Alliance Manuscript #LSA-2021-01209-TRR

Dr. Kai Sun
The University of Texas Health Science Center at Houston
1825 Pressler ST
Houston 77030

Dear Dr. Sun,

Thank you for submitting your Research Article entitled "Adipose tissue-specific ablation of *Ces1d* causes metabolic dysregulation in mice". It is a pleasure to let you know that your manuscript is now accepted for publication in Life Science Alliance. Congratulations on this interesting work.

DISTRIBUTION OF MATERIALS:

Again, congratulations on a very nice paper. I hope you found the review process to be constructive and are pleased with how the manuscript was handled editorially. We look forward to future exciting submissions from your lab.

Sincerely,
